# A Novel Method of Monitoring Surface Subsidence Law Based on Probability Integral Model Combined with Active and Passive Remote Sensing Data

**Rui Wang [1,2,\*]****, Kan Wu [1], Qimin He [3], Yibo He [4], Yuanyuan Gu [5] and Shuang Wu [6]**

1. School of Environment Science and Spatial Informatics, China University of Mining and Technology, Xuzhou 221116, China; 1604@cumt.edu.cn
2. College of Resources an Architecture, Gannan University of Science and Technology, Ganzhou 341000, China
3. School of Geography Science and Geomatics Engineering, Suzhou University of Science and Technology, Suzhou 215009, China; heqimin@usts.edu.cn
4. Institute of Land Reclamation and Ecological Restoration, China University of Mining and Technology, Beijing 100083, China; heyibo@student.cumtb.edu.cn
5. College of Surveying and Geoinformatics, Tongji University, Shanghai 200092, China; yuan1@tongji.edu.cn
6. Chinese Antarctic Center of Surveying and Mapping, Wuhan University, Wuhan 430079, China; wushuang1004@whu.edu.cn
* Correspondence: r-wang@cumt.edu.cn; Tel.: +86-134-7995-1198

**Abstract:** For the accurate and high-precision measurement of the deformation field in mining areas using different data sources, the probability integral model was used to process deformation data obtained from an Unmanned Aerial Vehicle (UAV), Differential InSAR (DInSAR), and Small Baseline Subset InSAR (SBAS-InSAR) to obtain the complete deformation field. The SBAS-InSAR, DInSAR, and UAV can be used to obtain small-scale, mesoscale, and large-scale deformations, respectively. The three types of data were all superimposed by the Kriging interpolation, and the deformation field was integrated using the probability integral model to obtain the complete high-precision deformation field with complete time series in the study area. The study area was in the WangJiata mine in Western China, where mining was carried out from 12 July 2018 to 25 October 2018, on the 2S201 working face. The first observation was made in June 2018, and steady-state observations were made in April 2019, totaling four UAV observations. During this period, the Canadian Earth Observation Satellite of Radarsat-2 (R2) was used to take 10 SAR images, the surface subsidence mapping was undertaken using DInSAR and SBAS-InSAR techniques, and the complete deformation field of the working face during the 106-day mining period was obtained by using the UAV technique. The results showed that the subsidence basin gradually expanded along the mining direction as the working face advanced. When the mining advance was greater than 1.2–1.4 times the coal seam burial depth, the supercritical conditions were reached, and the maximum subsidence stabilized at the value of 2.780 m. The subsidence rate was basically maintained at 0.25 m/d. Finally, the accuracy of the method was tested by the Global Navigation Satellite System (GNSS) data, and the medium error of the strike was 0.103 m. A new method is reached by the fusion of active and passive remote sensing data to construct efficient, complete and high precision time-series subsidence basins with high precision.

**Keywords:** land subsidence; InSAR; UAV; probability integration method

## 1. Introduction

With the rapid development of the national economy, energy demand continues to be strong, and coal consumption has been ranked first for primary energy. Large-scale coal mining damages the original structural stress of surrounding rocks, disturbs the groundwater system, and brings about serious geological disasters and ecological and environmental problems such as land subsidence, landslides, soil erosion, environmental

damage, and heavy metal pollution. The disasters caused by large-scale subsidence are increasing year by year [1,2].

Underground coal mining often causes surface subsidence, and gradually develops from low subsidence in a small area to large subsidence in a wide area, depending on the geological conditions, mechanical properties of rocks, and deployed mining methods [3]. Surface subsidence does not occur in a short period of time. Instead, its development needs a certain period, increasing as the mining face expands. Therefore, continuous monitoring is necessary. The large-scale subsidence caused by underground coal mining is a complex spatiotemporal process, and surface monitoring is of great significance for the prevention and control of geological hazards in mining areas. The most widely used technology for monitoring surface subsidence caused by coal mining is the conventional observation station method. For this method, a series of observation points are arranged every 5–25 m above the mining face and along its strike and dip. High precision leveling, total stations, and the GNSS stations are used for field monitoring at different mining stages of the working face to obtain the subsidence and deformation along the dip and strike of the coal seam through data processing. Such methods are based on point-by-point monitoring, with low spatial resolution and low efficiency. In long-term monitoring, it is difficult to ensure that the observation stations are not damaged due to the complex surface conditions in the mining area. In addition, it is difficult to protect the observation stations in the western mining areas.

With the development of modern measurement technology, the unmanned aerial vehicle (UAV) and interferometric synthetic aperture radar (InSAR) are reliable technologies for effective monitoring of large-scale regional subsidence. To monitor large subsidence in small areas, the UAV has the advantages of low cost and high efficiency, which effectively supplements the deficiencies of airborne geospatial data acquisition. However, the UAV has certain limitations in monitoring small ground subsidence, so it must be combined with other monitoring methods to improve the overall monitoring accuracy. The InSAR is an effective type of measurement and monitoring technology, which can achieve all-weather, wide-range, and long-time periodic continuous monitoring of surface subsidence. More importantly, archived data can be extracted to analyze the pattern before and after subsidence [4–6].

Affected by geological conditions and mining technology, large surface subsidence occurs in the western region when mining reaches a certain degree. Zhou et al. [7] used vertical photogrammetry technology to monitor the subsidence in the mining area. A maximum subsidence of 2.67 m was observed in a series of monitoring, and the error was 121 mm compared to the total station data. Research using vertical UAV photogrammetry and inclined photogrammetry proved that the error was approximately 100 mm. It can be applied in large subsidence monitoring in the mining area. However, in areas with smaller subsidence around the subsidence basin, it needs to be combined with other monitoring techniques for joint monitoring.

The differential InSAR (DInSAR) is an effective surface subsidence monitoring technique studied in recent years. It was fully verified through a large number of studies that the accuracy of the InSAR technique could reach centimeter or even millimeter scale [8,9]. This technology has been successfully deployed in monitoring studies of mining-related subsidence [10–12]. Based on the conventional InSAR technology, PS-InSAR [13,14], SBAS-InSAR [15,16], TCP-InSAR [17,18], and ITPA-InSAR [19] can accurately monitor small subsidence at the edge of the mining subsidence basin. Comparing the data collected from different levels, the monitoring accuracy can reach the millimeter scale. However, the monitoring accuracy of large subsidence in the basin was not high. In mining areas of Western China, surface subsidence is often complex and nonlinear, affected by various conditions, and large subsidence occurs within a short period of time during the mining process. Prior theoretical and practical research revealed that it was difficult to obtain high accuracy subsidence basin monitoring data by traditional InSAR technology alone

due to subsidence scale, vegetation cover, SAR image resolution, time interval, and other factors [20–22].

With further research on the InSAR technology, certain methods are available for large subsidence monitoring. R. Michel et al. [23] used the pixel-tracking method for surface monitoring of the Landers earthquake and obtained surface subsidence in the azimuth and range directions. Through evaluation and analysis, it was found that this method can be used for large subsidence monitoring. Subsequently, Strozzi et al. [24] used L-band JERS-1 images to monitor glacial movement in the Arctic region of Iceland through pixel-tracking and obtained a maximum cumulative offset of 6 m over 44 d. In mining area monitoring, Zhao et al. [25] used the offset-tracking method to monitor large surface subsidence in the Shangwan coal mine and Bulianta coal mine, which verified the feasibility of this method in large surface subsidence monitoring for the first time. In 2015, Fan et al. [26,27] combined the offset-tracking and phase superposition to obtain large surface subsidence above the working face in the Daliuta mining area. The results showed that the solution was effective. Wang et al. [28] used DInSAR, sub-band InSAR, and offset-tracking to jointly solve the regional subsidence in the mining area, and the feasibility of this fusion method in large subsidence monitoring was verified through comparison with GNSS data. Although the improved InSAR technique can monitor large subsidence, its accuracy is affected by the image resolution; also, for higher resolution, the cost increases. Moreover, it is influenced by the satellite itself, which has a fixed acquisition period and a poor flexibility. Tang et al. [29] combined GPS and InSAR to realize dynamic monitoring of the surface subsidence of the mining area, and obtained the dynamic surface subsidence contour map by fusion of the two sets of data. Zhou et al. [30] adopted the Helmert Variance Component Estimation (HVCE) method and the Back-Propagation Neural Network (BPNN). GNSS and InSAR data were fused to obtain the 3D deformation of the mining area. Mukherjee et al. [31] adopted the convolutional neural networks method for SAR image noise processing, and proposed the "GenInSAR", which has been well applied.

Based on the above problems, this study proposed a probability integral model combined with the UAV, DInSAR and SBASInSAR technologies to obtain high accuracy and complete coverage of the deformation field. Both SBASInSAR and DInSAR can help to cover small-scale subsidence, while the UAV covers large-scale subsidence. The high-precision image of a subsidence basin was obtained by a probability integral model fused with multi-source data. This study took the 2S201 working face of Wangjiata mine in Inner Mongolia as the study area. Four UAV observations and 10 SAR images Radarsat-2 (R2) were taken before and after the mining. The length of the working surface was about 1264 m. Ladder-shaped subsidence happened after 105 days of mining, and the maximum subsidence was 2.780m. We verified the feasibility of this method by comparing the data obtained by GNSS and the fusion method.

## 2. Materials and Methods

### 2.1. Study Area and Data

The study area is in the WangJiata mining area, which belongs to Xineng Mining Co., Ltd., located in Ejin Horo Banner, Erdos City, Inner Mongolia Autonomous Region. The mining area was an irregular polygon ranging 14.40 km from north to south, with a width of 6.77 km from east to west. The elevation ranged from 1150m to 1420 m, and the surface fluctuated greatly. The mining area is a vast coalfield, rich in resources and with an excellent coal quality. The depth of coal seam ranges approximately from 150 to 456 m. The specific geographical location is shown in Figure 1.

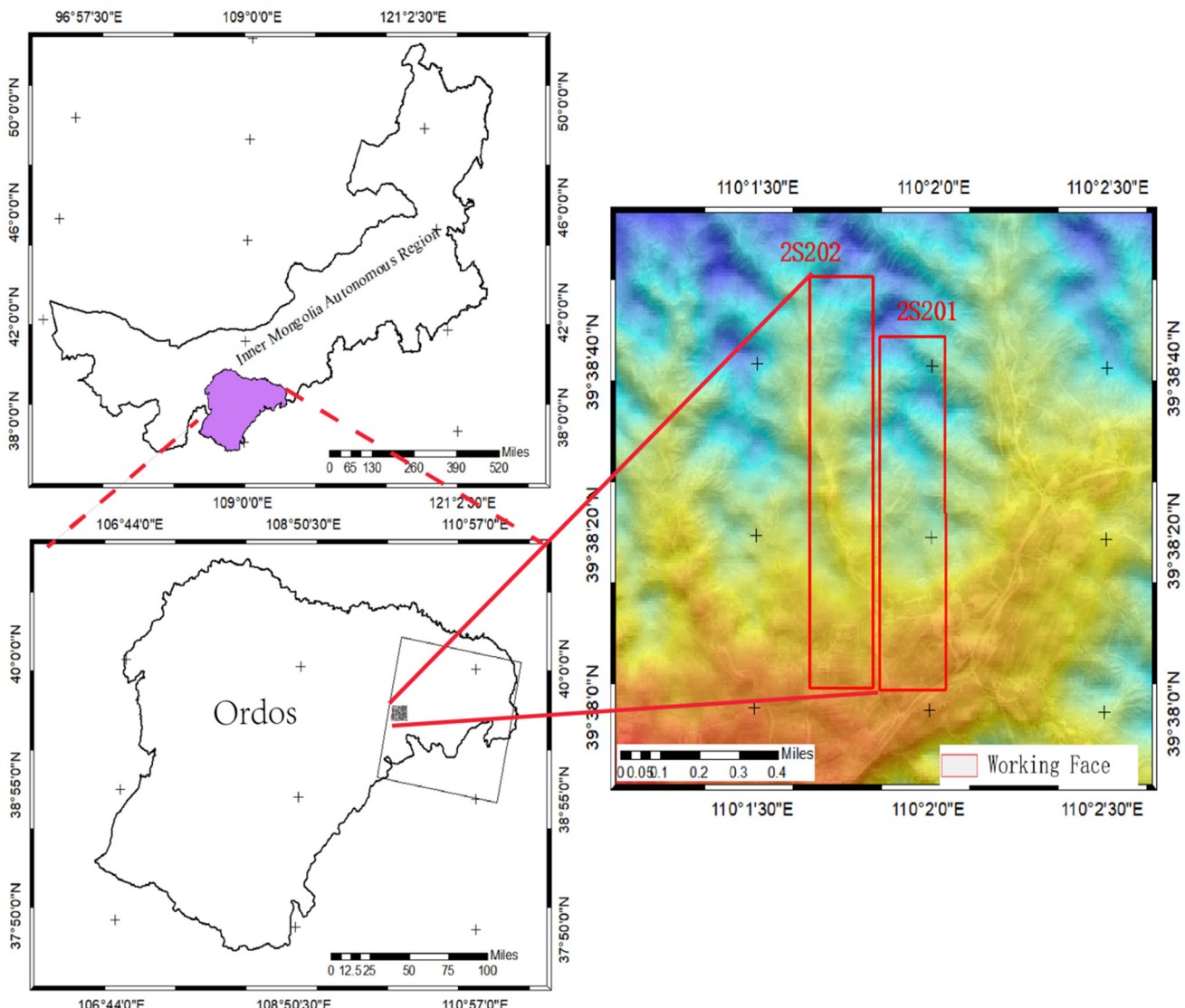

**Figure 1.** Location of the study area in WangJiata, Inner Mongolia Autonomous Region, China and photographs of building crack, collapse pit and ground crack caused by coal mining activity; 2S201 is the monitoring working surface.

On a long-term perspective, large-scale mining of underground coal resources often induces surface subsidence with different amplitudes. The surface soil in the mining areas of Western China is relatively loose and thick; thus, landslides and mudslides can be easily triggered. This causes ecological degradation and is a serious threat to life and property.

In this paper, UAV images and radar interferometric images were mainly used as the experimental solution data. UAV images were collected on-site using a Trimble UX5 UAV aerial survey system equipped with a SONY A5100 SLR camera [32]. The route height was set as 230 m when considering the terrain of the 2S201 working face and the height of the surrounding high voltage tower. The course overlap and the side overlap were 80%. Radar images were acquired using the RadarSat-2 radar satellite launched by the Canadian Space Agency (carrying a C-band sensor with a revisit period of 24 d) to collect the image data for differential interferometry. The acquisition dates and associated parameters of the interferometric images are shown in Tables 1 and 2.

**Table 1.** Optical image acquisition parameters.

| No. | UAV | Camera | Course Overlap% | Lateral Overlap% | Row Height | Collection Date |
|---|---|---|---|---|---|---|
| 1 | | | | | | 9 June 18 |
| 2 | Trimble UX5 | SONY A5100 | 80 | 80 | 23 | 4 September 18 |
| 3 | | | | | | 16 October 18 |
| 4 | | | | | | 16 April 19 |

**Table 2.** Parameters of RadarSat-2 images over WangJiata mine area.

| No. | Product | Beam Model | Polarization | Resolution/(m) (Rng × Az) | Acquisition Date | Pixel Center Lat-Lng (°) | Mean Incident Angle (°) |
|---|---|---|---|---|---|---|---|
| 1 | | | | | 9 June 18 | 39.5841–110.5944 | 35.2230 |
| 2 | | | | | 27 July 18 | 39.5852–110.5950 | 35.2232 |
| 3 | | | | | 20 August 18 | 39.5851–110.5961 | 35.2224 |
| 4 | | | | | 24 November 18 | 39.591–110.5977 | 35.2128 |
| 5 | SLC | Wide Multi-look Fine | HH | 2.6 × 2.4 | 11 January 19 | 39.5892–110.5969 | 35.2129 |
| 6 | | | | | 4 February 19 | 39.5627–110.5903 | 35.2124 |
| 7 | | | | | 28 February 19 | 39.5729–110.5952 | 35.2165 |
| 8 | | | | | 24 March 19 | 39.5899–110.5995 | 35.2207 |
| 9 | | | | | 17 April 19 | 39.5880–110.5955 | 35.2223 |

*2.2. Probability Integration Method*

The probability integral model takes the random medium theory as theoretical basis, and integrates both shift and distortion. The rock mass is regarded as a granular random medium, and the internal stress of the rock mass is destroyed so that the granular medium moves and deforms until the internal stress reaches equilibrium again. The process of equilibrium can be considered as a random process obeying statistical laws, which can be used to study the movement of rock mass and surface. From the statistical point of view, the whole mining unit can be composed of many tiny units. Thus, in the mining process, the subsidence of the rock mass and surface is equal to the sum of subsidence of each micro unit relative to the rock mass and surface. In the random medium theory, the movement, deformation, and subsidence of the rock mass and surface caused by the mining of tiny units demonstrate normal distribution and are consistent with the probability density distribution. The probability integral model is shown in Equation (1).

$$
\begin{aligned}
W(x,y) = W_0 \times \frac{1}{2}\left\{\left[\mathrm{erf}\left(\sqrt{\pi}\tfrac{x}{y}\right)+1\right] - \left[\mathrm{erf}\left(\sqrt{\pi}\tfrac{x-l}{r}\right)+1\right]\right\} \\
\times \frac{1}{2}\left\{\left[\mathrm{erf}\left(\sqrt{\pi}\tfrac{y}{r}\right)+1\right] - \left[\mathrm{erf}\left(\sqrt{\pi}\tfrac{y-L}{r}\right)+1\right]\right\} \\
r = \frac{H_0}{\tan\beta} \\
W_0 = \mathrm{mqcos}(\alpha)
\end{aligned}
\tag{1}
$$

where $W(x,y)$ is the subsidence at any point $(x,y)$ of the subsidence basin, $W_0$ is the maximum subsidence of the subsidence basin (where m is the average mining thickness, q is the subsidence factor, $\alpha$ is the coal seam dip), erf is the error function, $r$ is the mining influence radius (where $H_0$ is the average coal seam burial depth, $\beta$ is the mining influence angle), $l$ is the coal seam strike length, and $L$ is the coal seam dip length. After the coal seam is mined out, the original stress balance in the overburden is damaged. When the length and width of the void area exceed 0.2–0.5 times of the average mining depth, the movement, deformation, and damage of the rock mass around the working face gradually extend to the surface. This will eventually lead to discontinuous deformation of the surface, cave-in areas, landslides, and ground cracks, thereby damaging buildings, affecting the normal operation of roads and railways, affecting the phreatic layer, and inducing geological disasters and ecological damage [33–36]. The failure mechanism and evolution of overburdened rock mass are shown in Figure 2.

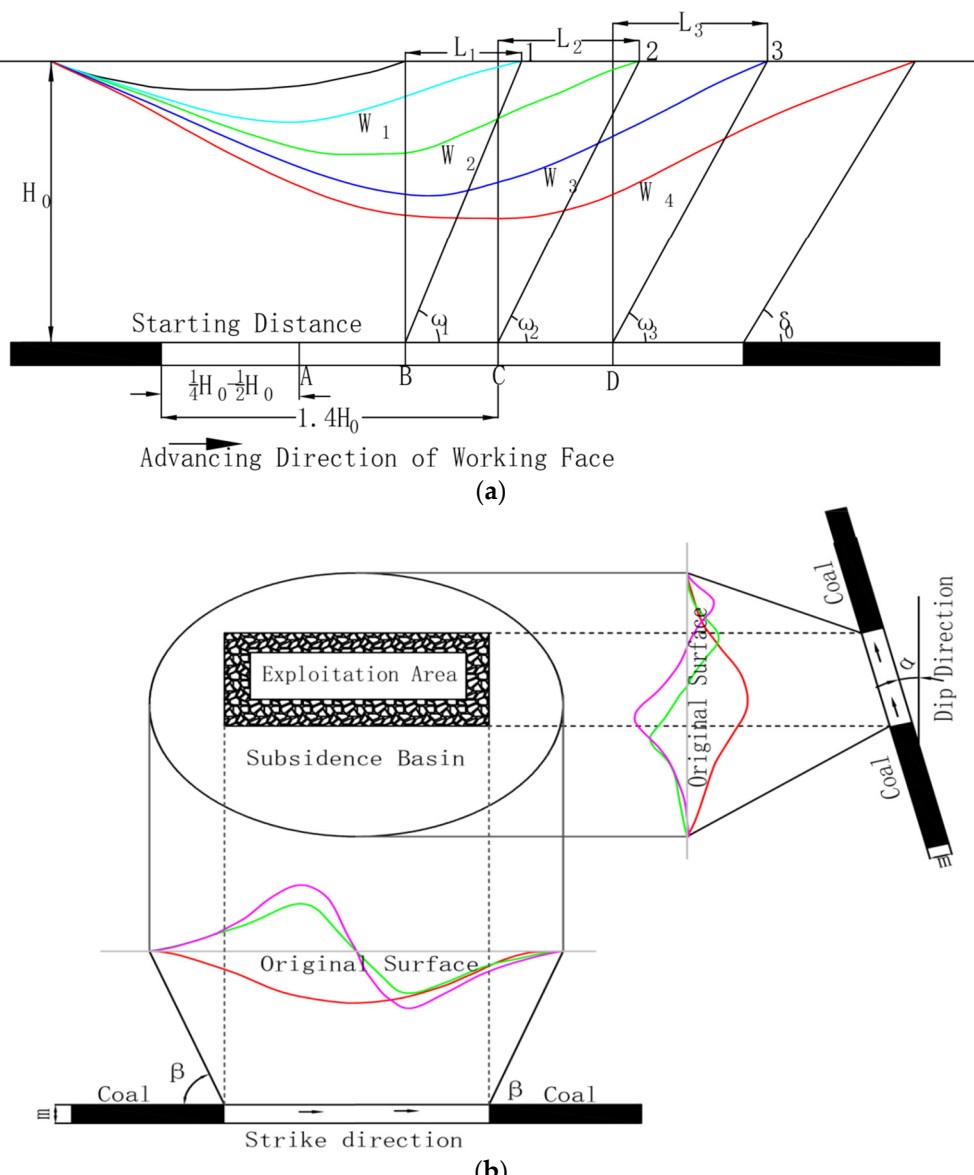

**Figure 2.** (**a**) Leading impact during the advancing process of the working face; (**b**) Relationship between surface movement basin and main section.

The surface subsidence caused by underground resource mining is a complex process. According to Figure 2a, the rock mass movement gradually begins to affect the surface when the mining face moves forward from the open cut position to point A on the main section. At this time, the distance from the open cut is 0.2–0.5 $H_0$, which is the starting distance. When mining continues to points B, C, and D, subsidence happens at points 1, 2, and 3 on the ground in front of the working face due to the influence of mining. $L_i$ is the influence distance, $W_i$ is the subsidence curve, and $\omega_i$ is the leading influence angle. The surface undergoes deformation in different directions when affected by coal mining. In Figure 2b, the red curve indicates subsidence, the green curve inclination, and the pink curve horizontal movement. The 2S201 working face is 1264 m long and 272 m wide, the subsidence factor is 0.79, the coal seam burial depth is 3.26 m, and the mining influence angle ($\tan\beta$) is 2.4.

## 2.3. Geometry Principle

With the development of InSAR technology, the application areas are expanding, including the measurement of phase difference between the satellite and the Earth's surface

through two satellite channels in the same region. The distance-based phase difference is mainly caused due to two reasons: (1) The difference in position between two or more satellite tracks, but the difference is much smaller than the distance from the satellite to the Earth's surface, so that it can be applied to topography deformation monitoring. (2) The observed regional displacement between two or more satellite acquisitions, which may be induced by exploitation of subsurface resources, earthquakes, and volcanic movements. Therefore, when the SAR system makes two or more observations of the same area, the geometric position of the area changes relative to the sensor. In other words, the surface is deformed. The technique for obtaining surface deformation by two or more interferometric measurements is called the DInSAR technique.

The reflected signal of the SAR system is affected by noise, topography, atmosphere, and surface movement during data acquisition. The radar interferometry principle and the geometric relationship of surface deformation are shown in Figure 3.

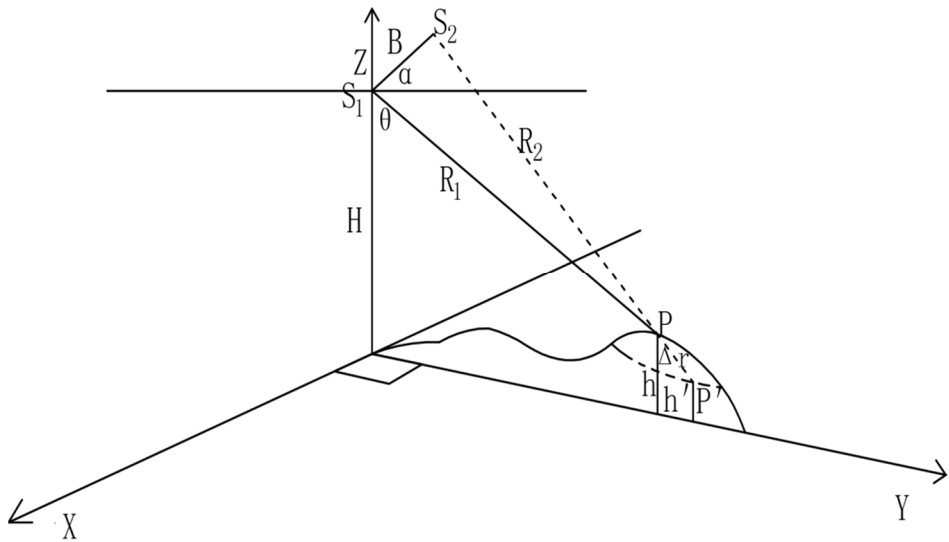

**Figure 3.** DInSAR principle and geometric relationship of surface deformation.

In Figure 3, $S_1$ and $S_2$ are two different shooting positions of the same sensor, $R_1$ and $R_2$ are the distance from the SAR sensor to the target point, $\theta$ is the incident angle, B is the baseline, $\alpha$ is the angle between the baseline and the horizontal direction, and $\Delta r$ is the LOS deformation of the target point. The interferometric phase of surface deformation is calculated as follows: the interferometric phase $\phi$ indicates that the same target $P$ is deformed to the point P' at two different acquisition times of the satellites $S_1$ and $S_2$. The interferometric phase $\phi$ is influenced by several factors and is calculated as follows [37]. In this experiment, two SAR images taken before and after the deformation and high precision DEM were used for interferometry. During the process of calculating, DEM was projected to the ground coordinate system of the base image. DEM was shifted into topography phrase $\phi_{Topo}$ after the projection, as shown in Equation (3).

$$\phi_{Int} = \phi_{Topo} + \phi_{Flat} +]\phi_{Defo} + \phi_{Atm} + \phi_{Noise} \tag{2}$$

$$\phi_{Topo} = -\frac{4\pi}{\lambda} \times \frac{B_{\perp}^0}{R_1 \sin \theta_0} h \tag{3}$$

$$\phi_{Flat} = -\frac{4\pi}{\lambda} B_{\parallel}^0 \tag{4}$$

$$\Delta r = \frac{\lambda}{4\pi} \phi_{Defo} = \frac{\lambda}{4\pi} (\phi_{Int} - \phi_{Flat} - \phi_{Topo}) \tag{5}$$

In Equation (2): $\phi_{Topo}$: phases influenced by topography; $\phi_{Flat}$: phase due to reference surface factors; $\phi_{Defo}$: phase due to surface deformation factors along the line of sight (LOS); $\phi_{Atm}$: phase due to atmospheric delay factors; $\phi_{Noise}$: phase due to noise factor. In Equation (3): $\lambda$ is the wavelength of the radar satellite, $R_1$ are the distances of the satellite when it passes the target $P$, $\sin\theta_0$ comes from the sensor of incident angle, $B_\perp^0$ represents the vertical baseline of projection and $h$ represents the ground elevation. In Equation (4), $B_\parallel^0$ is the projection of the base line from the ground point to the satellite line at the ellipsoidal reference point. In Equation (5), $\Delta r$ represents deformations.

The research of He et al. [38] showed that the vertical baseline will not be zero when DInSAR is used; therefore, the accuracy of DEM plays a key role in the solution of the two-pass method, as shown in Equation (6)

$$\sigma\Delta R = \frac{B_\perp^0}{R\sin\theta_0}\sigma h \tag{6}$$

In Equation (6): $\sigma\Delta R$ is the solution error of the shape variables; $\sigma h$ is the DEM elevation error.

In order to improve the accuracy of the ALOS DEM, the DEM obtained by the drone is integrated with it to improve the accuracy. The principal method is to calculate the standard deviation of the DEM obtained by ALOS DEM and UAV, respectively, and then calculate the weight of the two fusion DEMs according to the standard deviation [36,39], and conduct data fusion according to the weight. The fusion model is shown in Equations (7) and (8), and the fusion result is shown in Figure 4.

$$P = \frac{s_2{}^2 - ls_1s_2}{s_1{}^2 + s_2{}^2 - 2ls_1s_2} \tag{7}$$

$$h_r = Ph_{y1} + (1-P)h_{y2} \tag{8}$$

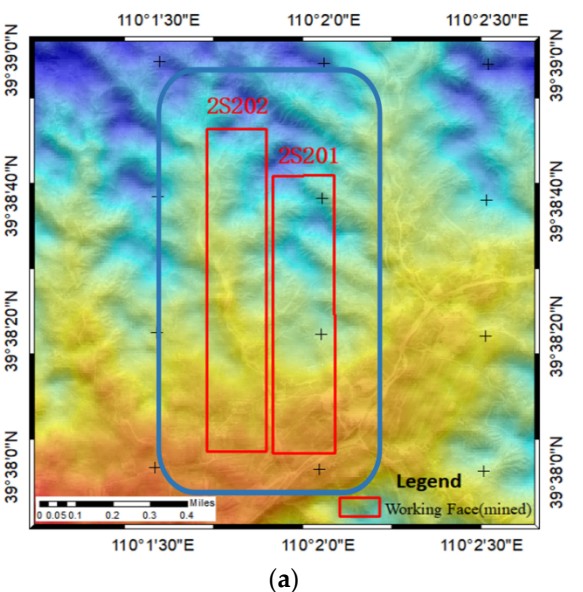 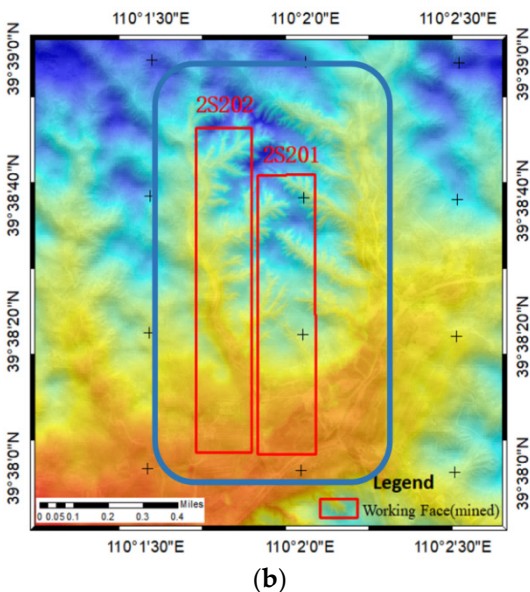

(**a**)  (**b**)

**Figure 4.** (**a**) ALOS DEM; (**b**) Fused DEM; 2S201 and 2S202 are working faces, the data is of production.

In Equation (6), $P$ is the weight of data fusion, and its value ranges from 0 to 1; $s_1$ and $s_2$ are standard deviations of fusion DEM; $l$ is the correlation coefficient of fusion image error; In Equation (7), $h_r$ is the DEM data after fusion while $h_{y1}$ and $h_{y2}$ are the two DEM data before fusion.

Figure 4 shows the area resolution and elevation in the blue range have been greatly improved.

SBASInSAR was used to identify the boundary in the subsidence basin. Time sampling rate was added into small baseline subset which realized further deformation measurement of high space density. Using the SBASInSAR, N+1 SAR images of the same area were obtained in chronological order. One of these images was chosen as base image and was matched to other SAR images. M differential interferometry images were generated by N+1 SAR images. After phase unwrapping, absolute correction was made to all images of differential interferometry through a stable area or a reference pixel with known deformation. Therefore, SAR data can be used to improve the time sampling frequency of deformation measurement and space coverage of the studied area. SBASInSAR, with stable algorithm, was applied to all pixels with high coherence within the coverage of the image. Due to the limitation of errors caused by the inaccurate DEM and the elimination of atmospheric effects, the result of high coherence regions is in scale of millimeters.

The conventional continuous interferometry method embedded inside the GAMMA software was used to process valid images. In 1992, Zebker et al. [40] found that the deformation of unit pixel should be half of the wavelength to accurately and effectively calculate the deformation in the direction of radar LOS. The wavelength of RadarSat-2 is 0.056 m, and the maximum theoretical deformation per unit pixel that can be solved by the interferogram is 0.028 m. When the deformation of adjacent unit pixels between any two images is greater than half of the wavelength, the deformation may not be correctly analyzed in the time domain.

*2.4. UAV Subsidence Monitoring*

Mining areas in Western China have complex terrain with more gullies, valleys, and weaker rocks. The surface soil layer is thicker, and the depth of the first coal layer is relatively shallow. Therefore, the surface subsidence caused by mining is larger, and it is difficult to monitor the surface with conventional methods. The UAV can supplement or strengthen conventional surveying and mapping methods. It is more flexible than the existing aerial photogrammetry or satellite imaging, with small relative error [41]. In addition, it has the advantages of fast speed, high efficiency, low cost, and high temporal resolution to obtain high spatial resolution images [42].

The UAV photogrammetry data is mainly processed by the aerial triangulation method. Aerial triangulation refers to continuous overlapping aerial images, combined with a small number of image control points in the field to establish the route model or area network model corresponding to the field by means of photogrammetry. In the encryption of air triangulation, position and orientation system data and ground image control points were used for direction. The principle of this method is based on the condition that the projection center point, image point, and corresponding ground image control point are collinear. The single image is taken as the solution unit, and the beam of each image is connected into an area by combining the image points of the same name and the field control points between the images. An overall adjustment was carried out to calculate the coordinates of the encryption points. The basic theoretical formula is the co-linear Equation (9) for the central projection. Two corresponding error equations can be listed by the image point coordinates of each photograph. The six pending parameters of the outer orientation elements of each image, i.e., the three spatial coordinates of the photographic site and the three independent

orientation parameters in the beam rotation matrix, can be solved according to the least squares criterion for leveling, and finally, the coordinates of each encryption point.

$$
\begin{aligned}
x - x_0 &= -f \frac{a_1(X - X_S) + b_1(Y - Y_S) + c_1(Z - Z_S)}{a_3(X - X_S) + b_3(Y - Y_S) + c_3(Z - Z_S)} \\
y - y_o &= -f \frac{a_2(X - X_s) + b_2(Y - Y_S) + c_2(Z - Z_S)}{a_3(X - X_S) + b_3(Y - Y_S) + c_3(Z - Z_S)} \\
a_1 &= \cos\varphi\cos\kappa - \sin\varphi\sin\omega\sin\kappa \\
a_2 &= -\cos\varphi\sin\kappa - \sin\varphi\sin\omega\cos\kappa \\
a_3 &= -\sin\omega\cos\omega \\
b_1 &= \cos\omega\sin\kappa \\
b_2 &= \cos\omega\cos\kappa \\
b_3 &= -\sin\omega \\
c_1 &= \sin\varphi\cos\kappa + \cos\varphi\sin\omega\sin\kappa \\
c_2 &= -\sin\varphi\sin\kappa + \cos\varphi\sin\omega\cos\kappa \\
c_3 &= \cos\varphi\cos\omega
\end{aligned}
\tag{9}
$$

In Equation (9): $(x,y)$ are the image plane coordinates; $(x_0,y_0)$ are the principal point coordinates of the image; $f$ is the image principal distance; $(X,Y,Z)$ are the ground coordinates of the object; $(X_S,Y_S,Z_S)$ are the coordinates of the camera station in the ground coordinate system; $(\varphi,\omega,\kappa)$ are the outer azimuth angles of the image; $(a_i,b_i,c_i)$ are the directional cosine represented by the outer azimuth angles. The flow chart for the encryption of aerial triangulation of UAV images is shown in Figure 5.

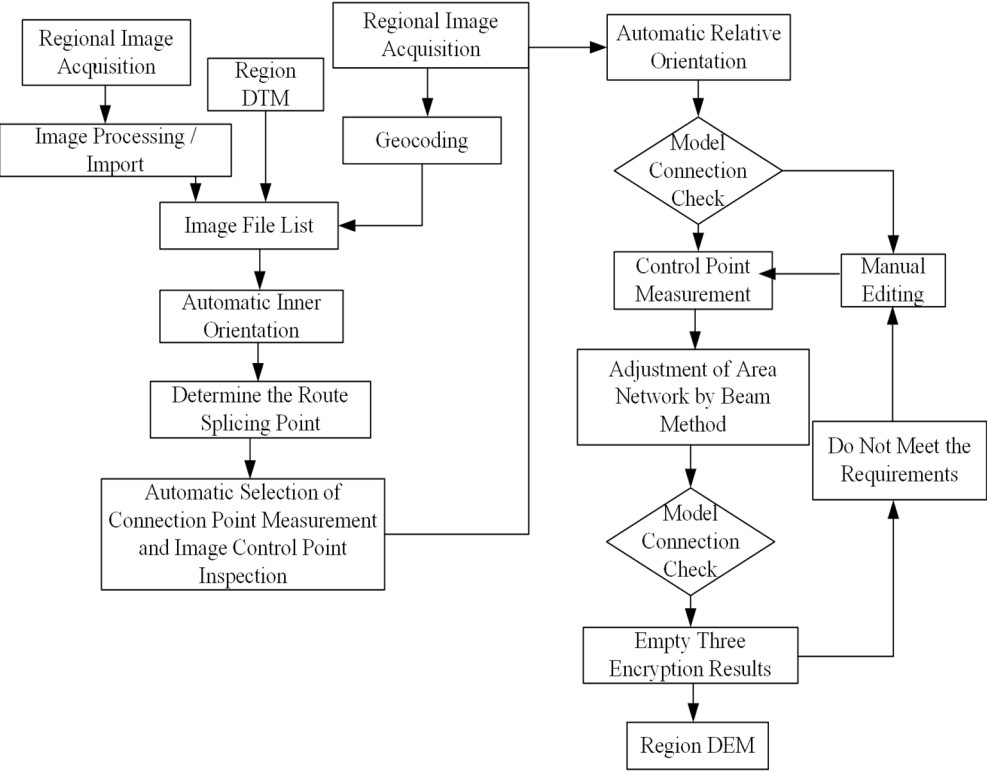

**Figure 5.** Flow chart of aerial triangulation encryption and DEM generation.

In the encryption of aerial triangulation, the average elevation of the area of data collection was introduced to improve the solution accuracy. In the eastern region, the terrain was relatively gentle, and the average elevation was generally calculated based on the DEM of SRTM [43] or the elevation of ground control points. In the western region, the terrain was hilly and undulating, so the average elevation calculation affected the accuracy of the solution. Therefore, in this paper, the DTM model of the survey area was introduced

in the solution process instead of the average elevation to improve the accuracy of aerial triangulation encryption points.

Four UAV monitoring observations were used from the beginning of mining in the WangJiata coal mine to six months after the end of the extraction. For each acquisition, the same flight parameters and ground resolution were used. The orthophotos, line drawings, point clouds, and DEMs were obtained for each phase of the 2S201 working face through image interpretation. The actual changes of working surface before and after mining can be obtained effectively through orthophotos and DEMs, which provide a reliable technical means to study the large subsidence pattern of the surface. Many scholars have studied the impacts of flight height, camera parameters, image control point accuracy, and ground resolution of the generated DEM accuracy. Zhou et al. [7] studied the accuracy of UAV-generated DEMs for the WangJiata 2S201 working face and found the reference accuracy was approximately 0.096 m.

*2.5. Data Fusion Method*

With the development of modern measurement technology, the monitoring of surface subsidence caused by mining has gradually changed from a traditional single method to a modern multi-method combination. The fusion of data collected by each method to improve whole subsidence basin modeling accuracy is a hot research topic in recent years. In this paper, both UAV and InSAR were mainly used to monitor large subsidence in the mining areas of Western China.

According to the "coal pillar and mining regulations of buildings, water, railways and main roadways", proposed by the China National Coal Association, the edge of the surface deformation induced by the mining of underground coal resources when there is a 10 mm subsidence, and the subsidence area greater than 10 mm is the mining-affected area. When the subsidence ranged from $-0.028$ m to 0 m, data of UAV and cumulative deformation of DInSAR were ignored, while the data of SBASInSAR were retained. When the subsidence ranged from $-0.096$ m to $-0.028$ m, data of UAV and SBASInSAR were ignored, while the data of cumulative deformation of DInSAR were retained. When the subsidence ranged from $-0.096$ m to $-0.028$ m, data of UAV and SBASInSAR were ignored while the data of cumulative deformation of DInSAR were retained. When the subsidence was less than $-0.096$ m, data of cumulative deformation of DInSAR and SBASInSAR were ignored, while the data of UAV were retained. Then, three deformation fields were superimposed into one complete deformation field by kriging interpolation.

Ordinary kriging is a basic method in geo-statistics for finding optimal and unbiased interpolation estimates. When $Z(x)$, the regionalization variable, is second order law, the basin $V(x_0)$ with $x_0$ as the center is estimated. The linear estimator can be used for estimation. The equation is as follows:

$$Z_K^* = \sum_{\alpha=1}^{n} \lambda_\alpha Z_\alpha \tag{10}$$

In Equation (10), $\lambda_\alpha$ is the weight coefficient of $Z_\alpha$ representing the weight of $\alpha(1, 2, 3, \ldots, n)$ deformation values $Z_\alpha$ to the estimated $Z_K^*$. In the interpolation calculation using the ordinary kriging method, the kriging equation set was listed to solve for the full coefficients of $\lambda_\alpha$. Secondly, the kriging variance was listed. The equation is as follows:

$$\begin{cases} \sum\limits_{\alpha=1}^{n} \lambda_\alpha \overline{\gamma}(v_\alpha, v_\beta) + \mu = \overline{\gamma}(v_\alpha, V) \\ \sum\limits_{\alpha=1}^{n} \lambda_a = 1 \ (\alpha = 1, 2, 3, \cdots, n) \end{cases} \tag{11}$$

In the kriging interpolation process, it is necessary to calculate the variation value of the theoretical function. Equation (11) can be used to find the estimated value $\gamma^*(h)$ of the theoretical variogram of $\gamma(h)$.

$$\gamma^*(h) = \frac{1}{2N(h)} \sum_{i=1}^{N(h)} [Z(x_i) - Z(x_i + h)]^2 \tag{12}$$

In Equation (12): $\gamma^*(h)$ is the theoretical function variance value; $Z(x_i)$ is the monitoring value at subsidence point $i$; $Z(x_i + h)$ is the monitoring value with a distance of h away from $Z(x_i)$ spatially; $N(h)$ denotes the number of sample data point pairs with a spatial separation distance of $h$. Using a spherical model, 15 search points are set, influence range is set to 20 m, search radius is set to 20 m, and empirical values are taken as other parameters. The flow chart of the fusion method is shown in Figure 6.

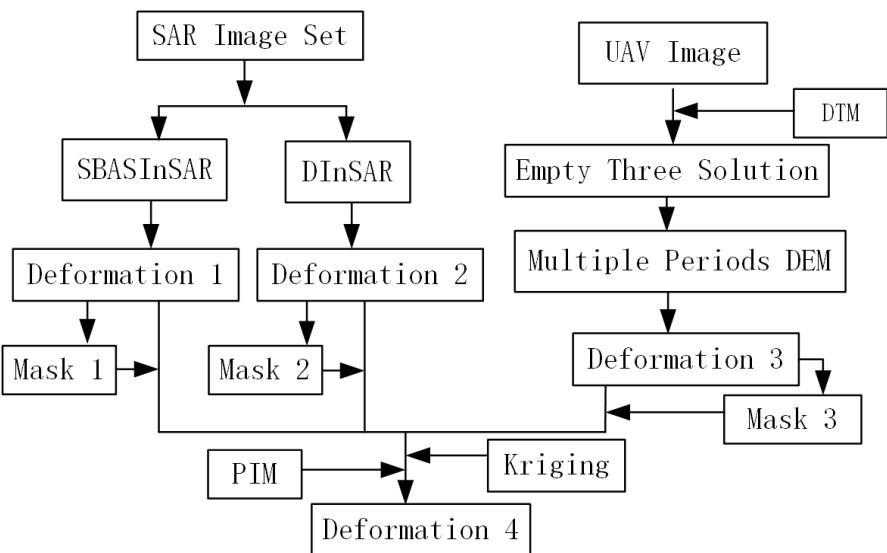

**Figure 6.** Flow chart of the fusion method.

## 3. Results

### 3.1. Data Fusion Result

The mining of the 2S201 working face in the WangJiata mining area started on 12 July 2018, and finished on 25 October 2018. The total length of the working face was 1264 m, and the mining lasted for 105 d. The surface environment of the 2S201 working face is complex, and the terrain was undulating. The surface soil is loose sandy soil; thus, there are numerous rain-washed gullies. Ground monitoring is difficult using traditional methods. Therefore, this study proposed the construction of a high-precision subsidence basin using a probability integral model combined with multi-source data to analyze the subsidence pattern. This paper focuses on the vertical deformation. The deformation in the LOS direction is converted to the vertical direction by the incident angle of the SAR sensor, and then the deformation in the vertical direction obtained by the UAV is integrated to study the overall deformation law.

The subsidence map in Figure 7a was based on the subsidence data obtained by the three methods of SBAS-InSAR, DInSAR, and UAV. Reliable data were obtained by mask processing according to the subsidence monitoring accuracy of the three methods, and finally, the initial subsidence results were obtained through superposition and kriging interpolation. Figure 7b was obtained by secondary fusion of the data in Figure 7a. According to Figure 6, it can be concluded that the probability integral model can not only fill in the voids of Figure 7a but also filter out the noise points. Finally, a more complete and smooth subsidence map with higher accuracy was obtained.

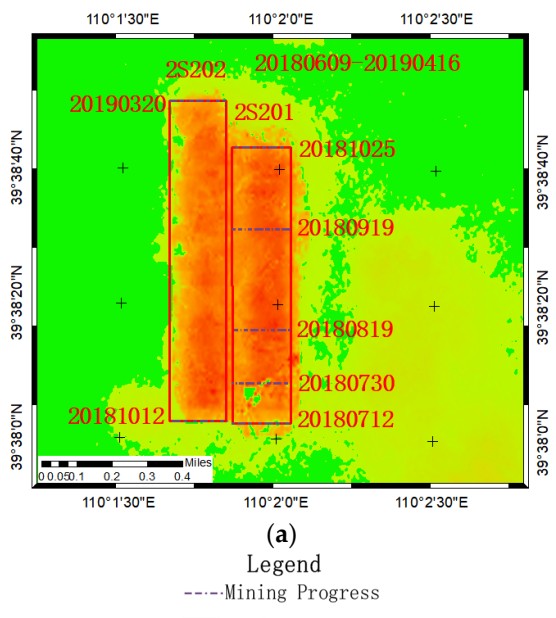
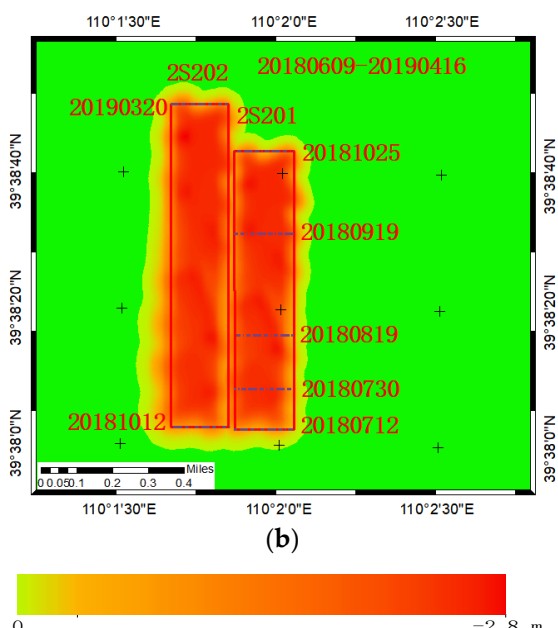

**Figure 7.** Example of fusion methods. (**a**) Interpolation and superposition subsidence map by SBASInSAR, DInSAR and UAV; (**b**) subsidence map by the fusion method; The date marked in red is the mining time; 2S201 and 2S202 are working faces; the data is of production.

### 3.2. UAV and InSAR Result

In the experiment, two techniques, UAV and InSAR, were used to interpret the data of surface subsidence. The large-scale deformation was obtained by UAV image calculation, and the small-scale subsidence was obtained by SAR image calculation.

Figure 8 show the subsidence maps obtained by the UAV method. The UAV images in Figure 8 can be processed to obtain large subsidence with a maximum subsidence of 2.760 m. However, the small-magnitude subsidence around the basin and the edge of the subsidence basin cannot be identified. The images show higher noise due to the gullying phenomenon caused by rainwater scouring ofthe ground with looser soils above and around the working face.

Figure 9 shows the time series subsidence maps between two adjacent images of both InSAR methods. Figure 8 show the subsidence maps obtained by the InSAR method. Influenced by the residual subsidence of mining at the 2S101 working face on the east side of the 2S201, subsidence occurred on the surface of the 2S101 working face from 9 June 2018 to 11 January 2019. The subsidence gradually decreased over time. Subsequently, the surface subsidence gradually disappeared and finally stabilized. The mining of the 2S201 working face officially started on 12 July 2018. As the working face advanced, the surface began to deform. The subsidence expanded along the direction of the working face advancement. From 12 July 2018 to 25 October 2018, the strength of the mining workwas 12 m per day. The SAR images of surface subsidence were out of coherence, and only small subsidences could be identified. As the surface subsidence gradually decreases with the completion of mining, the InSAR technique can identify the edge of the subsidence basin. The maximum subsidence of 0.144 m was monitored from 24 November 2018 to 11 January 2019.

Figure 10 shows the time series cumulative subsidence maps for both InSAR methods from 12 July 2018 to 17 April 2019. Figure 10 shows the cumulative subsidence maps obtained by the InSAR method. The maximum cumulative subsidence identified in the sight direction during this time period was 0.342 m. As the working face advanced and time progressed, the subsidence basin gradually expanded due to the influence of mining. Figure 10a–c show the time period of mining, during which the amount of surface

subsidence was larger, and SAR images were severely out of coherence. Thus, a smaller amount of subsidence was identified.

Figure 11 shows the cumulative surface subsidence map obtained by the UAV. These figures suggest that the UAV can identify the area with larger subsidence. The maximum cumulative subsidence was 2.760 m. According to these two methods, the surface subsidence basin extended along the mining direction of the working face. From the initial mining to full mining, the subsidence value increased continuously, and the maximum subsidence point was kept behind the stop line.

Figure 12a shows the cumulative subsidence profile of the working face from the probability integral model combined with InSAR and UAV data and the advancing position of the working face in the direction of the strike. Figure 12b shows the 3D view of the subsidence basin of the 2S201 working face as of 17 April 2019. At the early stage of mining, the surface subsidence and subsidence range gradually increased, and the surface subsidence showed a funnel shape. Approximately 275.8 m was mined, which was over 1.4 times the average mining depth on 9 August 2018. The flat bottom appeared at the center of the subsidence basin, and the maximum subsidence stabilized. It can be seen from Figure 11b that from south to north, the subsidence was slightly different with the different mining depths, and the flat bottom of the whole basin was not completely horizontal. The scope of the subsidence basin can be analyzed from the projection contour. At the end of mining, as the advancement rate slowed, the surface subsidence basin was still slowly expanding along the mining direction. However, the subsidence value and subsidence rate were reduced.

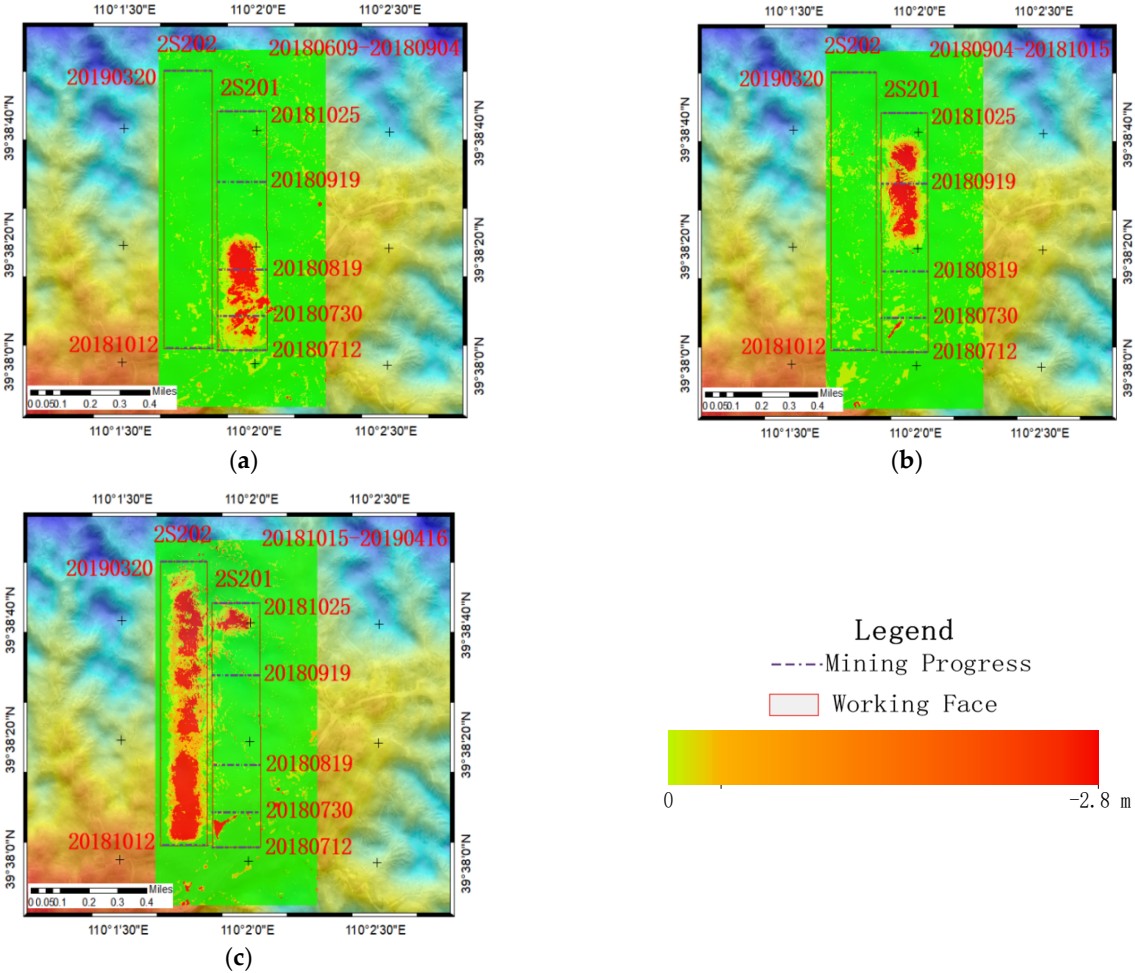

**Figure 8.** (**a**–**c**) are respectively the settlement maps of UAV time series; 2S201 and 2S202 are working faces; the data is of production.

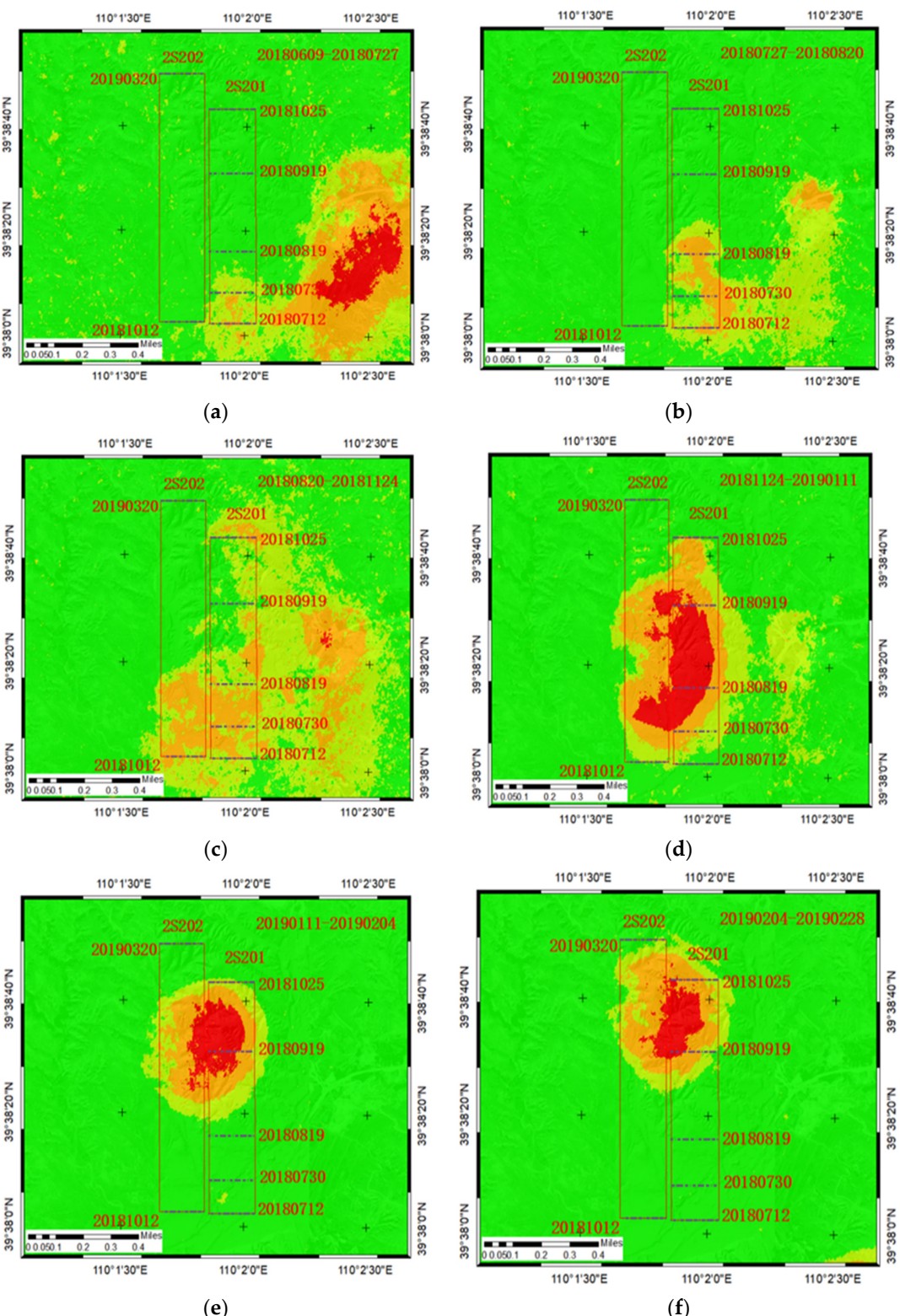

**Figure 9.** *Cont.*

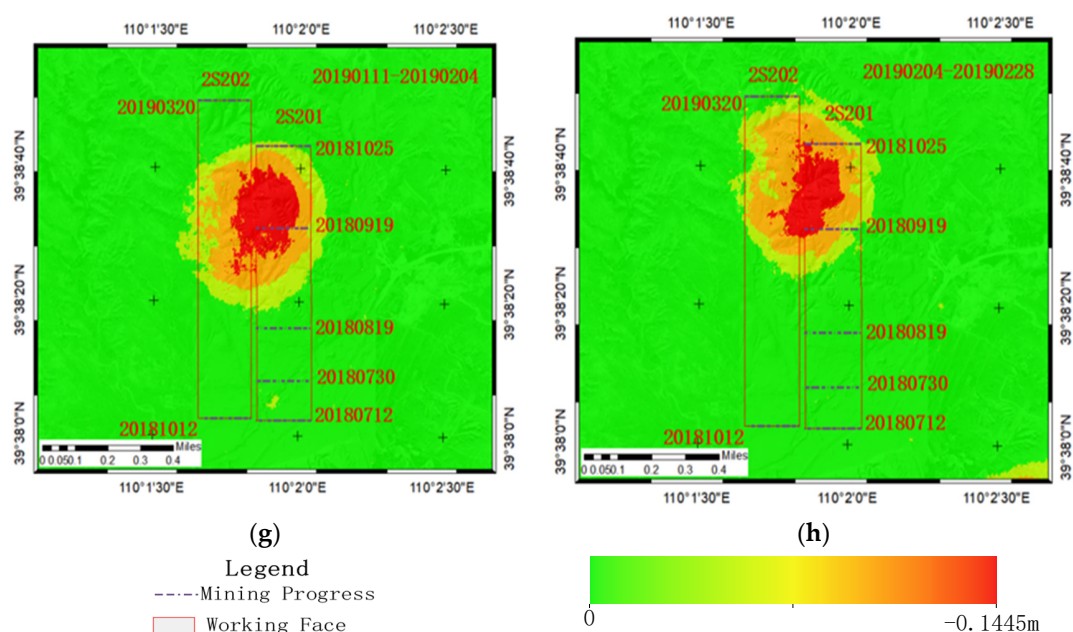

**Figure 9.** (**a**–**h**) are respectively the settlement maps of DInSAR.

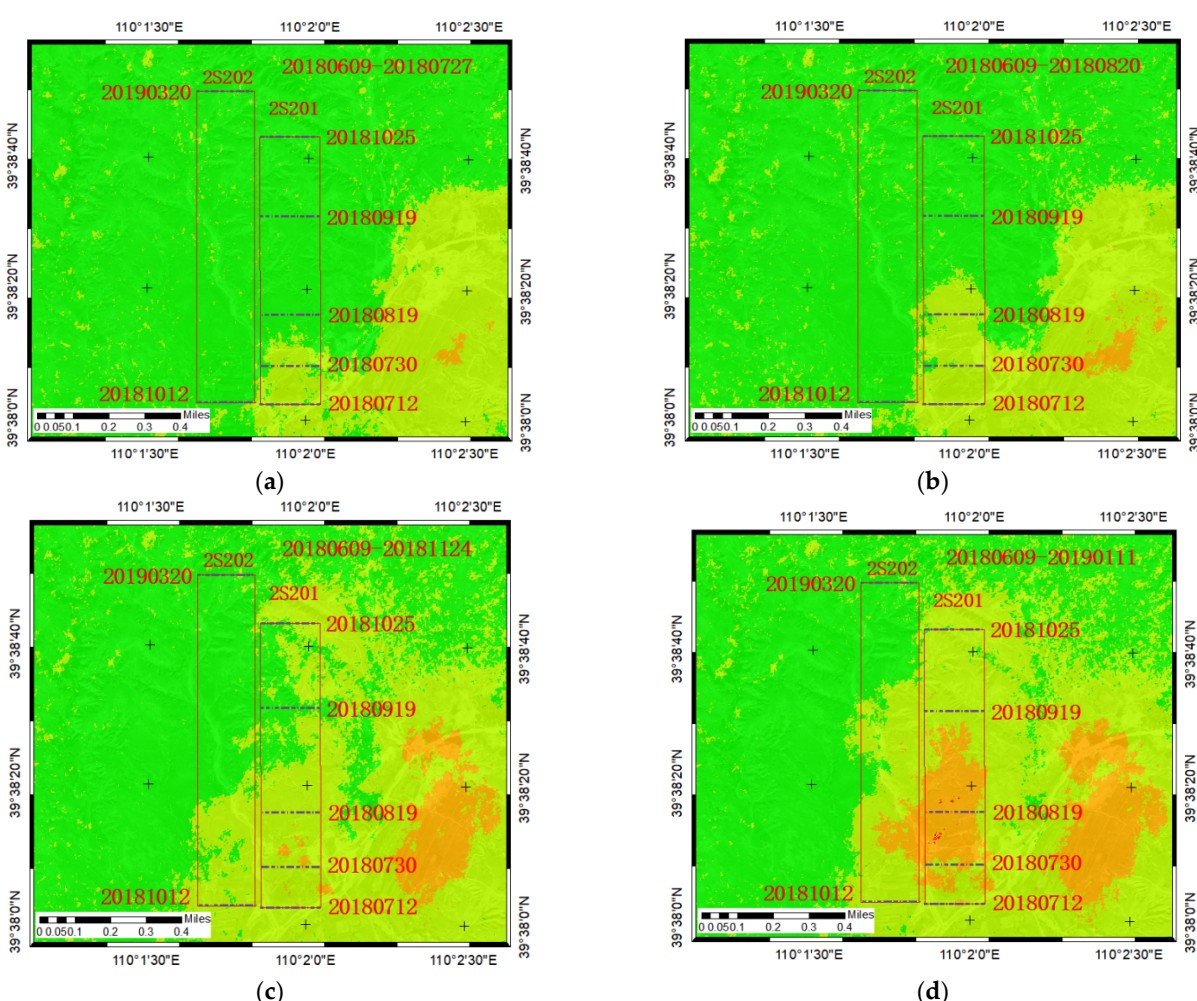

**Figure 10.** *Cont.*

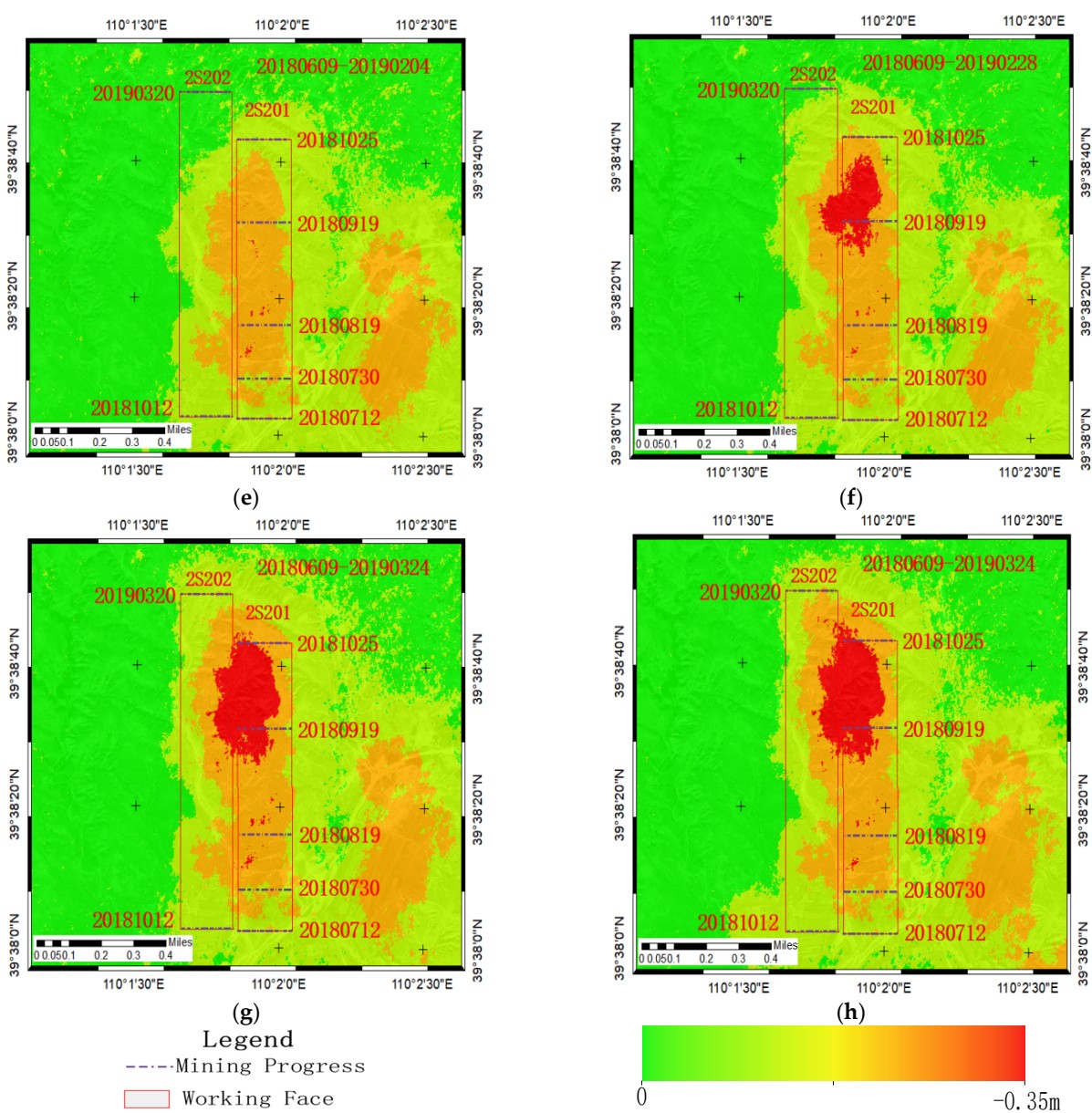

**Figure 10.** (**a**–**h**) are respectively the accumulated settlement maps of DInSAR.

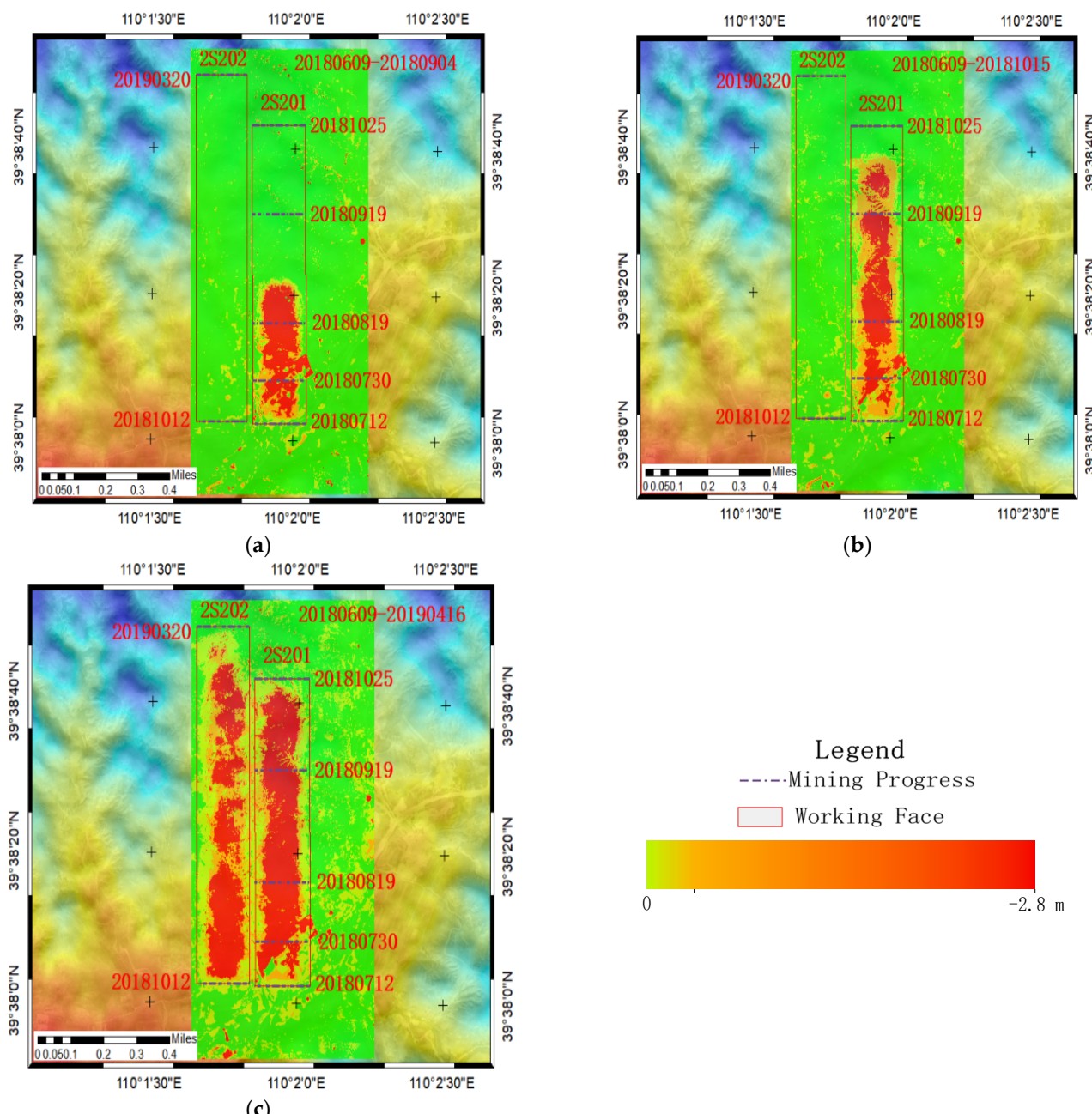

**Figure 11.** (**a**–**c**) are respectively the accumulated settlement maps of UAV time series; 2S201 and 2S202 are working faces; the data is of production.

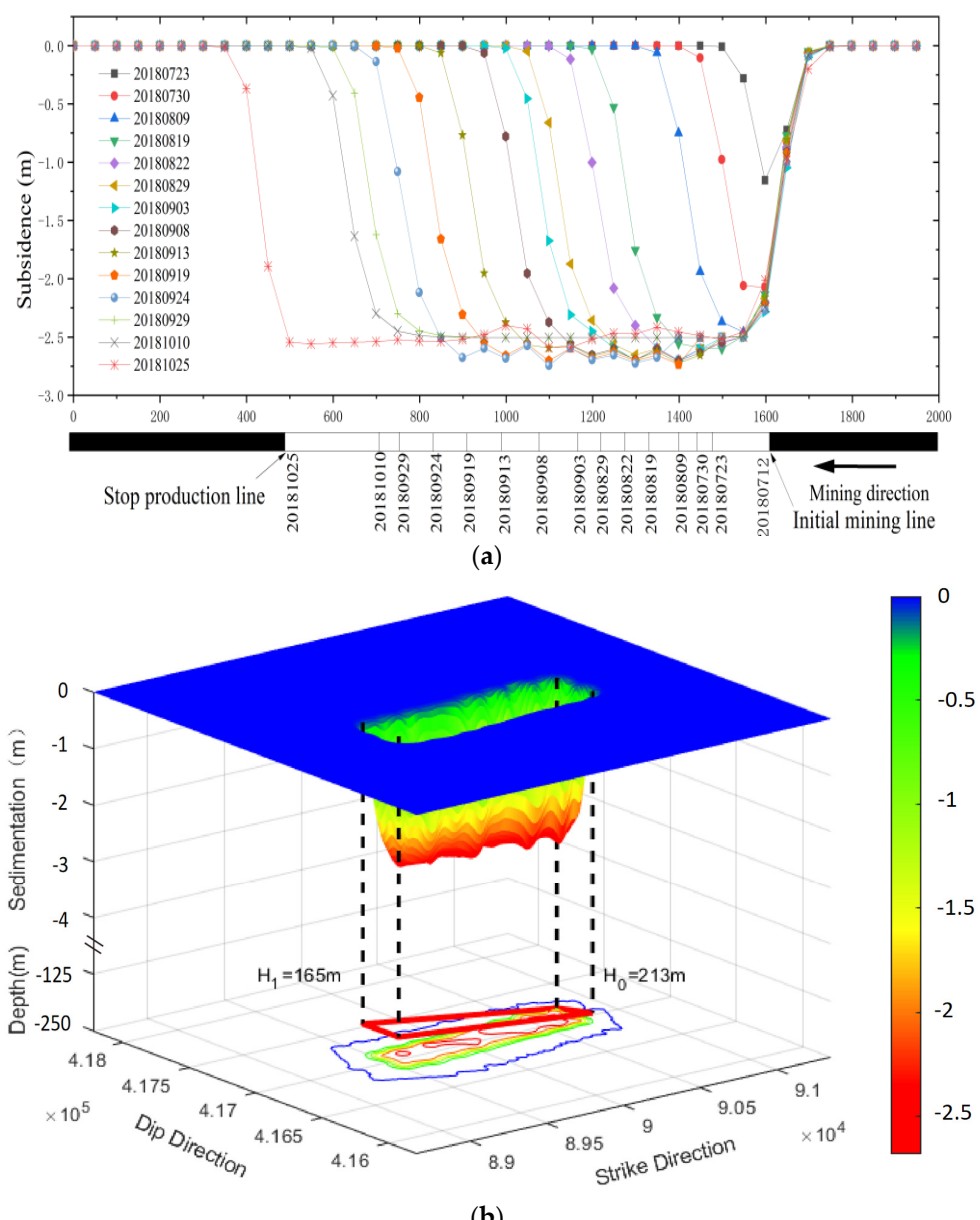

**Figure 12.** Dynamic cumulative curve of the 2S201 working face, locations of working face during advancement in strike direction, and 3D subsidence map. (**a**) is the time-series cumulative subsidence curve of 2S201 working face; In (**b**), the red box indicates the working face, strike direction means the north, dip direction means the east-west, The corresponding scale is the coordinate value.

## 4. Discussion

### 4.1. Comparative Analysis of Data from InSAR, UAV, and GNSS

Cumulative surface subsidence maps from 12 July 2018 to 16 April 2019, were obtained using InSAR, UAV, and fusion methods, as shown in Figure 13.

Figure 13a shows the cumulative subsidence obtained by the InSAR method, Figure 13b shows the cumulative subsidence obtained by the UAV method, and Figure 13c shows the cumulative subsidence obtained by the fusion method. The maximum cumulative subsidence measures from the three methods were 0.342 m, 2.760 m, and 2.682 m, respectively. The cumulative subsidence obtained by the InSAR method was the smallest, but the edge of the subsidence basin can be identified. Although the UAV can identify larger subsidence in the subsidence basin, it cannot identify small subsidence at the edges of the basin. Thus, it cannot accurately locate the edges of the basin. Due to the complex surface

conditions, the deployed subsidence pattern analysis points failed to identify the maximum subsidence. Combined with high-resolution UAV images, the maximum subsidence of the basin was determined as 2.760 m. In this study, we proposed combining the InSAR and UAV monitoring methods to give full play to the advantages of both methods so as to obtain a high accuracy subsidence basin in the whole mining area.

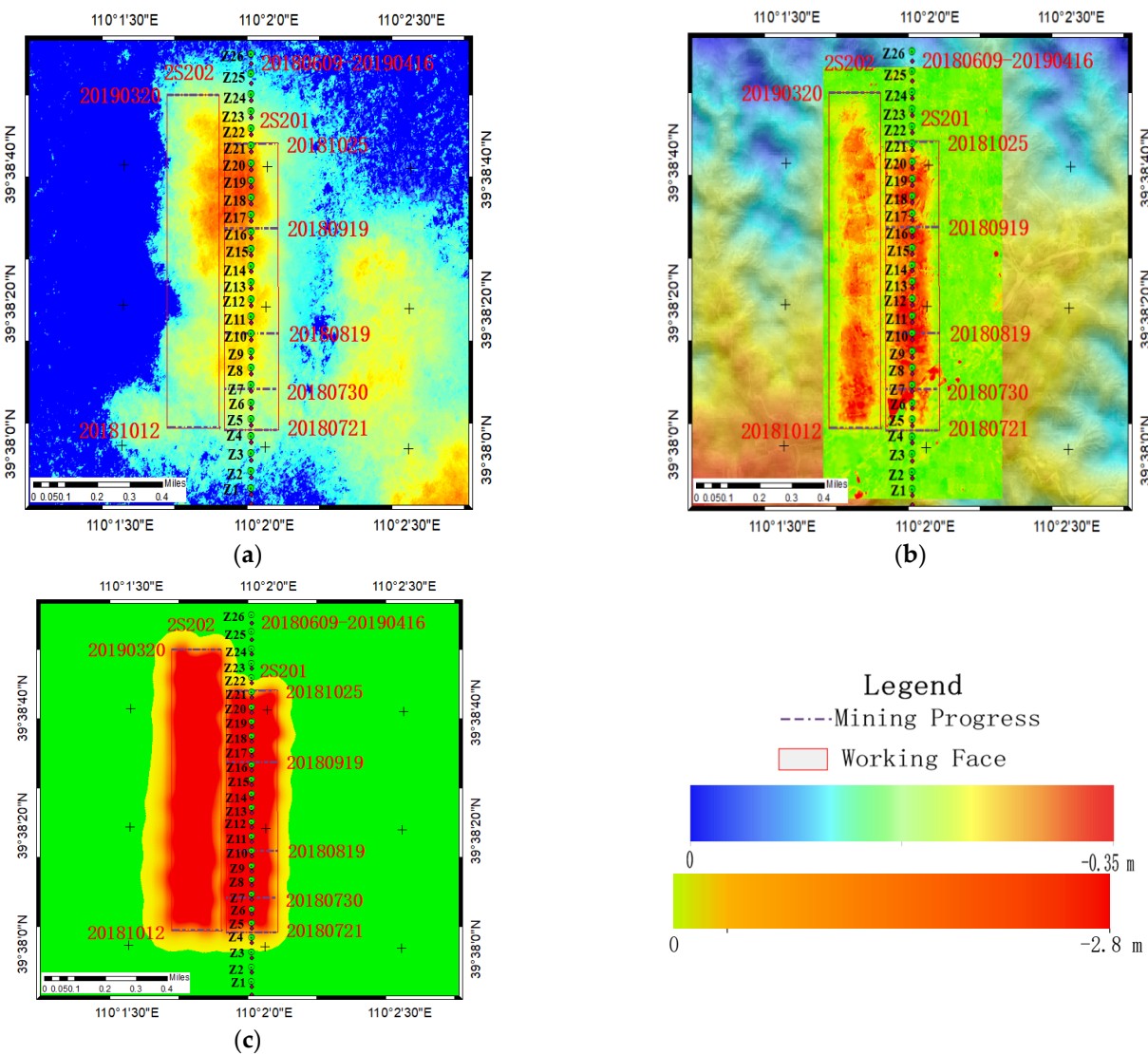

**Figure 13.** Cumulative subsidence obtained by three methods: (**a**) cumulative subsidence by InSAR; (**b**) cumulative subsidence by UAV; (**c**) cumulative subsidence by fusion method; Z1 to Z26 are GNSS monitoring point; 2S201 and 2S202 are working faces; the data is of production.

Eleven GNSS points were randomly selected from the subsidence monitoring points, and the subsidence values were compared with the subsidence values obtained by the fusion method to verify the accuracy, as shown in Table 3. The medium error of the selected points was 0.103 m. Among them, the error at point 10 was larger, with a value of 0.206 m.

As can be seen from Table 3, the probability integral model combined with InSAR and UAV methods has good performance in constructing large-scale subsidence basins. Most of the single-point errors were within 0.2 m. The median error obtained by InSAR alone is 1.426 m, with 0.099 m for UAV, and 0.103 m for fusion method. The results show that the median error of the UAV is close to the fusion method and slightly better than the fusion method. The traditional InSAR technique with low accuracy, cannot identify large-scale subsidence, but it is sensitive to small-scale subsidence and can identify the boundary of

the subsidence basin. The UAV method can identify large-scale subsidence, but not the boundary of the subsidence basin. Although the accuracy of the fusion method is slightly lower than that of the UAV method, the boundary of the subsidence basin can be identified.

**Table 3.** Comparison of subsidence values between measured and fusion methods.

| No. | InSAR (m) | UAV (m) | Fusion (m) | GNSS (m) | InSAR/GNSS (m) | UAV/GNSS (m) | Fusion/GNSS (m) |
|-----|-----------|---------|------------|----------|----------------|--------------|-----------------|
| 1 | −0.051 | −0.042 | −0.187 | −0.174 | −0.123 | −0.132 | 0.013 |
| 2 | −0.062 | −1.158 | −1.409 | −1.304 | −1.242 | −0.146 | 0.105 |
| 3 | −0.106 | −1.297 | −1.495 | −1.388 | −1.282 | −0.091 | 0.107 |
| 4 | −0.152 | −2.52 | −2.628 | −2.542 | −2.39 | −0.022 | 0.086 |
| 5 | −0.098 | −0.096 | −0.154 | −0.111 | −0.013 | −0.015 | 0.043 |
| 6 | −0.113 | −0.475 | −0.561 | −0.507 | −0.394 | −0.032 | 0.054 |
| 7 | −0.034 | −0.091 | −0.156 | −0.077 | −0.043 | 0.014 | 0.079 |
| 8 | −0.131 | −0.01 | −0.284 | −0.164 | −0.033 | −0.154 | 0.12 |
| 9 | −0.178 | −2.131 | −2.529 | −2.323 | −2.145 | −0.192 | 0.206 |
| 10 | −0.150 | −2.646 | −2.724 | −2.668 | −2.518 | −0.022 | 0.056 |
| 11 | −0.047 | −1.817 | −1.925 | −1.857 | −1.81 | −0.04 | 0.068 |
| 12 | −0.087 | −1.235 | −1.291 | −1.146 | −1.059 | 0.089 | 0.145 |
|  | **Medium Error** |  |  |  | 1.426 | 0.099 | 0.103 |

The errors at individual points were relatively large due to special reasons. For areas with special topographic and geological conditions, the observation stations are easily damaged and cannot be used for a long period. In other cases, the largest subsidence point was not in the observation line due to special reasons. This paper provided a reliable method to solve the above problems and to systematically study the subsidence law of the working face.

*4.2. Analysis of Observation Method and Subsidence Law*

The 2S201 working face is 1264 m long and 272 m wide. With a flight area of about 1.719 million square meters, the UAV takes about 60 min to observe large-scale surface subsidence in a short time. Figure 10 shows that UAV technology can effectively monitor large-scale surface subsidence areas, and high spatial resolution orthography obtained by UAV technology can effectively identify surface cracks, providing reliable and practical data for evaluating damage caused by coal mining. InSAR can identify a subsidence range of more than 10 mm on the surface, fully combined with the advantages of UAV in constructing a complete subsidence basin with high efficiency, high precision and high resolution, and overcome the defect of traditional methods to obtain subsidence data for parameter inversion.

Based on monitoring and fusion data, the subsidence evolution of the 2S201 working face is analyzed. The rock hardness of the 2S201 working face was below average, and the surface soil layer, which belonged to sandy land was thick and soft. Therefore, the subsidence rate from the edge to the center of the subsidence basin was fast, due to stepped subsidence. With the advancement of the working face, the maximum subsidence rate and the advancement rate had a certain relationship. As the advancement rate went up, the maximum subsidence rate increased. When the advancement rate reached a certain value, the maximum subsidence rate stabilized. The subsidence rates of the 2S201 working face from 12 July 2018 to 25 October 2018, are shown in Figure 14, and the maximum cumulative subsidence values are shown in Figure 15.

The mining distances, advancement rate, and subsidence rates at different times during the mining of the 2S201 working face are shown in Figure 11. From 12 July 2018 to 9 August 2018, the 2S201 was not fully mined. In this period, as the advancement rate increased, the subsidence rate also increased. From 9 August 2018 to 10 October 2018, the working face was fully mined, and the subsidence rate was unaffected by the advancement rate, which stabilized at 0.25 m/d. The period between 10 October 2018 to 25 October 2018 represented the end of mining. The mining distance was large, and the subsidence rate

gradually decreased. From Figure 11, it can be seen that the subsidence rate was mainly affected by the advancement rate and whether the working face was fully mined or not. The subsidence rate remained the same during the supercritical period.

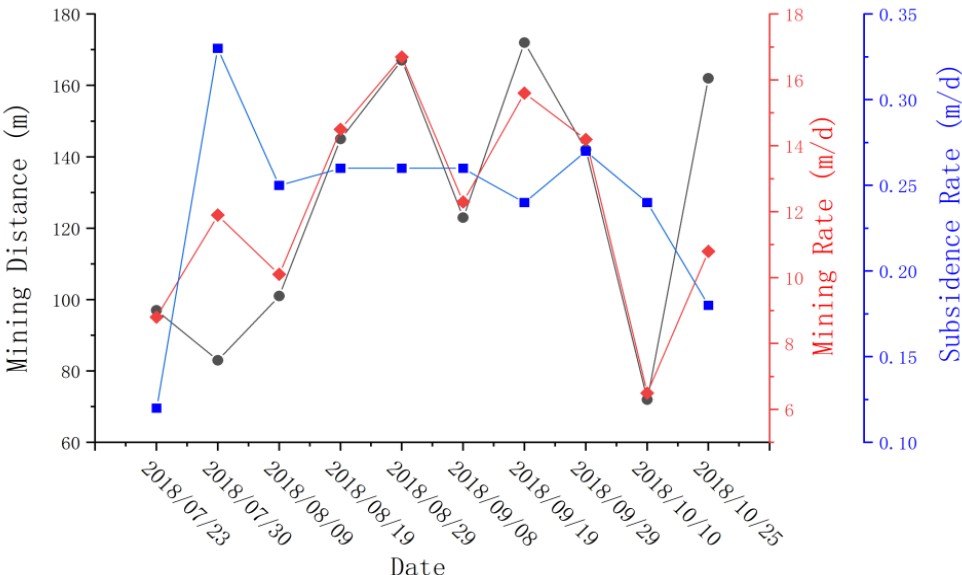

**Figure 14.** Time series subsidence rate graph. The black vertical axis represents the time node mining distance; the red vertical axis represents the time node average advancement rate; and the blue axis represents the time node average subsidence rate.

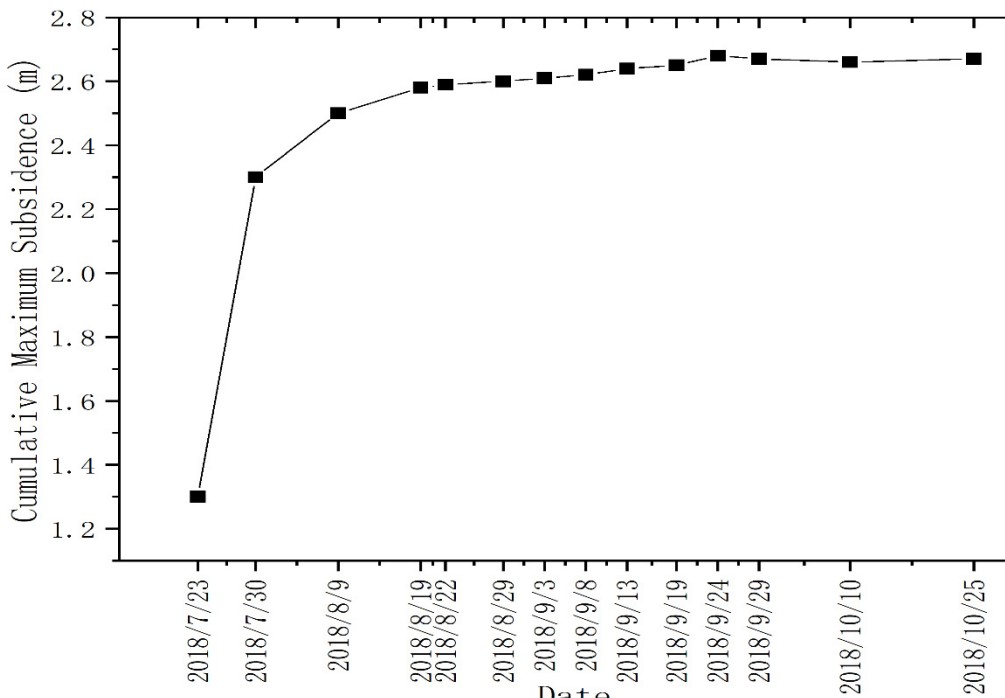

**Figure 15.** Time series maximum cumulative subsidence plot.

The recorded dates during the advancement of the working face and the corresponding cumulative subsidence values are shown in Figure 12. The cumulative maximum subsidence increased rapidly from 1.319 to 2.257 m from 12 July 2018 to 30July 2018. During the same period, the working face advanced 180 m. The cumulative maximum surface subsidence increased to 2.462 m from 30 July to 9 August 2018, with the working face

advancing by 101 m. According to the fusion method, the region with the maximum cumulative subsidence from 9 August to 25 October 2018, was stable. In the probability integral model, the maximum subsidence was 2.682 m, and the UAV monitored the maximum subsidence of 2.780 m. According to the mining subsidence theory, full mining is reached when the length and width of the mining face reaches 1.2–1.4 times the average mining depth. Then, the maximum subsidence of the subsidence basin no longer increases as the working face advances. The working face advanced 281 m from 12 July 2018 to 9 August 2018, and the mining width of the working face was 271 m. The average mining depth was 195 m; thus, it can be considered that underground mining has reached full mining conditions. As a result, the working face continued to advance from 9 August 2018 to 25 October 2018, but the maximum surface subsidence was essentially stable. After the mining of the working face was finished, the surface continued to deform and the subsidence basin continued to expand in a certain area in the mining direction of the working face. The subsidence value decreased to the edge of the basin.

## 5. Conclusions

In this study, a probability integral model combined with SBAS-InSAR, DInSAR, and UAV methods was used to extract the surface subsidence caused by the mining of underground coal to construct a complete and high-precision surface subsidence field of the WangJiata 2S201 working face in Inner Mongolia from 12 July 2018 to 16 April 2019. The results showed that the surface subsidence pattern during the time period could be effectively analyzed by the probability integral model fusing InSAR and UAV methods when there were few ground observation stations affected by the special conditions of the surface. The main conclusions of this study were as follows: (1) The standard deviation of the heterogeneous DEM is calculated, and the fusion weight model of the two DEM is built by the standard deviation, so as to improve the accuracy of DEM calculation. (2) The maximum cumulative subsidence from the initial mining to the completion of working face mining was 2.780 m. The experimental data and GNSS data were compared, and the medium in the direction of the working face advancement was 0.103 m, and the maximum error was 0.206 m. (3) Before the supercritical was reached, the rate of subsidence increased as the advancement rate increased. Furthermore, the subsidence rate remained the same when full mining was reached, and the cumulative subsidence graph showed that the region with the maximum subsidence was stable when full mining was reached. In summary, in areas where the surface environment is complex, traditional stations are vulnerable to damage and cannot be monitored for long time series, and the decoherence of SAR images is significant; the probability integral model fusing InSAR and UAV methods provides a solution to monitor areas with large subsidence while ensuring high accuracy. With the combination of UAV and InSAR, the subsidence basin data can be obtained more flexibly, and the fusion of this two types of data provides a new idea and method for studying the subsidence laws of mining areas. UAV orthophotography will be used in future studies to extract surface cracks and assess surface vegetation destruction, thereby providing some data support for mining ecological monitoring and assessment data.

**Author Contributions:** R.W.: Conceptualization, methodology, formal analysis, writing—original draft, writing—review and editing; K.W.: conceptualization, methodology, writing—review and editing; Q.H.: conceptualization, methodology, formal analysis; Y.H.: methodology, data curation, formal analysis; Y.G.: methodology, data curation, formal analysis; S.W.: methodology, data curation, formal analysis. All authors have read and agreed to the published version of the manuscript.

**Funding:** This work was supported by the Science and Technology Project of Jiangxi Education Department (No. GJJ191594), Mass Entrepreneurship and Innovation Project in Jiangxi Province (202113434002), Ningxia Key R & D Project (No. 2020BFG03009), the National Natural Science Foundation of China (No.51604266) and the Natural Science Foundation of Jiangsu Province (BK20190642).

**Institutional Review Board Statement:** Not applicable.

**Informed Consent Statement:** Not applicable.

**Data Availability Statement:** Not applicable.

**Acknowledgments:** The authors would like to thank the anonymous reviewers and the editor for their constructive comments and suggestions for this paper. Thanks to Shiqiao Huang, Xinpeng Diao, Liang Li, Dawei Zhou, Wenna Wang, Zhicong Wu, Yong Yan, for their guidance in writing this paper.

**Conflicts of Interest:** The authors declare no conflict of interest. The funding sponsors had no role in the design of the study; in the collection, analyses, or interpretation of data; in the writing of the manuscript, and in the decision to publish the results.

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
