# Peer review of "A Novel Method of Monitoring Surface Subsidence Law Based on Probability Integral Model Combined with Active and Passive Remote Sensing Data"

_remotesensing, doi:10.3390/rs14020299_

Round 1
Reviewer 1 Report
- Please declare your novelty in the introduction session and also the conclusion
- Based on Figure 6., you have performed data fusion between SAR and another method. Why do you use results based on line of sight for spaceborne SAR? Because the value will differ if compared with the drone due to incident angle position.
- Figure 14 and 15, please replace with the higher resolution one
Author Response
Thank you for your comments. We revised them according to the experts' opinions, and the results are as follows:
Point 1:Please declare your novelty in the introduction session and also the conclusion. 

Response 1: Thank you for your constructive and helpful suggestion.We have stressed the novelty in both the introduction and the conclusion.we have highlighted it in red in the latest draft.
Point 2: Based on Figure 6., you have performed data fusion between SAR and another method. Why do you use results based on line of sight for spaceborne SAR? Because the value will differ if compared with the drone due to incident angle position.
Response 2: Thank you for your specific comments.The main purpose of fusion of SAR and other methods is to extract small - and medium-sized deformation in line of sight. Then, the line-of-sight data is converted into vertical component data according to the SAR satellite parameters.Finally, the data is fused with the large-scale deformation data obtained by UAV to obtain the complete subsidence basin data.
Point 3:Figure 14 and 15, please replace with the higher resolution one.
Response 3: Thank you for your specific comments.In the latest draft, we have modified Figures 14 and 15.we have highlighted them in red in the latest draft.

Reviewer 2 Report
The authors propose a method to improve subsidence monitoring by fusing multiple sources of remote sensing data. Overall, the paper is well written and except for some minor typos like "Moreover, it is influenced by the satellite itself, which has a fixED acquisition period and a poor flexibility"(Line 118-119), the flow of the paper and the extent to which the methods are described and the results discussed is sufficient for the reader to understand the authors' work. However, I have few comments regarding the comparison with existing methods and comprehensiveness of the literature survey that could help improve the manuscript and make it more suitable for publication. I list those points below along with the References (at the end of this review) that can help the authors improve their manuscript.
- While the authors mention that fusion approaches using different types of remote sensing data were successfully used in the past, (for example, reference# 28 in the paper, as described in Lines 113-117), in the results section of their manuscript, they do not compare their proposed method with any such method, irrespective of whether the authors use the exact same type of remote sensing data as those other methods. The question that needs to be explored here, is, how do we know qualitatively or quantitatively, that fusing the types of data that the authors used and/or their proposed method for fusing them is/are optimal for surface subsidence monitoring ? In the same vein, it would add some more context to the manuscript if the authors discuss some more related traditional and recent fusion based approaches like [1,2].
- The authors mentioned that part of the noise in their data gets eliminated during the fusion process, but it would be more interesting to add a discussion of the pros and cons of using recent data-driven noise filtering techniques like [3].
References
- T. Fuquan, C. Zuxi and W. Hanying, "Application of GPS/InSAR fusion technology in dynamic monitoring of mining subsidence in western mining areas," 2012 2nd International Conference on Consumer Electronics, Communications and Networks (CECNet), 2012, pp. 2420-2423, doi: 10.1109/CECNet.2012.6202159
- W. Zhou, W. Zhang, X. Yang, and W. Wu, “An Improved GNSS and InSAR Fusion Method for Monitoring the 3D Deformation of a Mining Area,” IEEE Access, vol. 9. Institute of Electrical and Electronics Engineers (IEEE), pp. 155839–155850, 2021 [Online]. Available: http://dx.doi.org/10.1109/ACCESS.2021.3129521
- S. Mukherjee, A. Zimmer, X. Sun, P. Ghuman and I. Cheng, "An Unsupervised Generative Neural Approach for InSAR Phase Filtering and Coherence Estimation," in IEEE Geoscience and Remote Sensing Letters, vol. 18, no. 11, pp. 1971-1975, Nov. 2021, doi: 10.1109/LGRS.2020.3010504
Author Response
Thank you for your comments. We revised them according to the experts' opinions, and the results are as follows:
Point 1: While the authors mention that fusion approaches using different types of remote sensing data were successfully used in the past, (for example, reference# 28 in the paper, as described in Lines 113-117), in the results section of their manuscript, they do not compare their proposed method with any such method, irrespective of whether the authors use the exact same type of remote sensing data as those other methods. The question that needs to be explored here, is, how do we know qualitatively or quantitatively, that fusing the types of data that the authors used and/or their proposed method for fusing them is/are optimal for surface subsidence monitoring ? In the same vein, it would add some more context to the manuscript if the authors discuss some more related traditional and recent fusion based approaches like [1,2]. 

Response 1: For the fusion approaches mentioned in reference# 28, high resolution SAR images must be used to ensure the accuracy of solution and fusion, further descriptions have been made in line 116 to 119 in the manuscript. Meanwhile, a new method mentioned in this paper, which fuse the active and passive remote sensing data so that high resolution SAR images are no longer needed. Also, passive remote sensing data can be obtained flexibly. This method not only saves costs, but also improves efficiency, providing a new way to construct subsidence basins formed on the surface due to underground coal mining. Some relative and latest fusion methods have been added in the Introduction. we have highlighted them in red in the latest draft.
Point 2: The authors mentioned that part of the noise in their data gets eliminated during the fusion process, but it would be more interesting to add a discussion of the pros and cons of using recent data-driven noise filtering techniques like [3].
Response 2: In the Introduction section, the content about the data noise filtering method is added.we have highlighted it in red in the latest draft.
Reviewer 3 Report
remotesensing-1503383-peer-review-v1
The manuscript “Novel Method of Monitoring Surface Subsidence Law Based on Probability Integral Model Combined with Active and Passive Remote Sensing Data” addresses an interesting and up-to-date subject, which adhere to Remote Sensing journal policies.
The manuscript tackles an interesting topic, related to the monitoring of subsidence in a mining perimeter by means of InSAR and UAV photogrammetry. Unfortunately, the manuscript falls short in some aspects and need additional work.
Some aspects that need improvement before resubmission:
- While there are good parts in the manuscript, the overall methodology does not have high novelty, there are many previous research related to data fusion from different sensors or methods of monitoring. I recommend to change the title a little bit, and also add InSAR and UAV in the title.
- Most likely the land survey was made with GNSS instrumentation, not GPS (which is the USA constellation of satellites). You should change that in the manuscript
- At R62 “…point-by-point monitoring, with low spatial resolution and low efficiency.”, GNSS measurements do have low efficiency, but it is incorrect to say “low spatial resolution”. Maybe a DEM derived from a few measured points has a low spatial resolution, but the individual point measurement has a good accuracy ≈2cm
- The whole UAV part of the article is severely underdeveloped. In order to monitor an area by photogrammetric and geomatics methods you must have a clear survey methodology and workflow. How did you plan the mission? How many GCPs and CPs did you place on the ground? How did you measure their position and with what instrumentation? What where the precisions (RMSE) obtained? What software did you use? How did you evaluate the subsidence between the UAV flights? In order to do this, a DEM of difference method must be used (e.g. geomorphic change detection). With UAV photogrammetry you can obtain DSM, but you mention DTM, how did you convert DSM to DTM.
- Regarding the last remark, an orthophoto or satellite image of the area of interest would be useful. Was there vegetation present (which can be a problem for both UAV and InSAR)?
- At R502 you mention the DTM of SRTM, but that has a very low resolution (approx. 30m) and can not be used in monitoring deformations
- The fusion can be further explained in the manuscript
- Discussion chapter needs improvements, with citations to similar studies
Author Response
Thank you for your comments. We revised them according to the experts' opinions, and the results are as follows:
Point 1: While there are good parts in the manuscript, the overall methodology does not have high novelty, there are many previous research related to data fusion from different sensors or methods of monitoring. I recommend to change the title a little bit, and also add InSAR and UAV in the title. 

Response 1: There are many researches on monitoring of surface subsidence in mining area, among which the combination of different methods and data fusion are both the research hotspots. The existing research results are mainly the combination of GNSS technology and InSAR technology and the combination of total station and InSAR technology, both of which are the combination of point and plane. Meanwhile, different InSAR technologies are used to compute and fuse high-precision SAR images. Few studies have been done on the fusion of optical data and radar data. In the early stage, we considered adding InSAR and UAV into the title, but after careful consideration, we believed that using “active and passive remote sensing data” could better represent the combination of the two technologies and the fusion of the two types of data.
Point 2: Most likely the land survey was made with GNSS instrumentation, not GPS (which is the USA constellation of satellites). You should change that in the manuscript.
Response 2: Thank you for your specific comments.We have revised the corresponding part in the article. The corrections have been highlighted in red.
Point 3: At R62 “…point-by-point monitoring, with low spatial resolution and low efficiency.”, GNSS measurements do have low efficiency, but it is incorrect to say “low spatial resolution”. Maybe a DEM derived from a few measured points has a low spatial resolution, but the individual point measurement has a good accuracy ≈2cm
Response 3:Thank you for your specific comments.The GNSS described in this paper has a low resolution, which is aimed at large-area deformation monitoring. GNSS can only perform single-point monitoring and cannot cover the entire area, which leads to its low spatial resolution. DEM can also be established if multi-point data acquisition is carried out in the subsidence area, but the efficiency is greatly reduced. And it is difficult to realize in the complex surface environment.
Point 4: The whole UAV part of the article is severely underdeveloped. In order to monitor an area by photogrammetric and geomatics methods you must have a clear survey methodology and workflow. How did you plan the mission? How many GCPs and CPs did you place on the ground? How did you measure their position and with what instrumentation? What where the precisions (RMSE) obtained? What software did you use? How did you evaluate the subsidence between the UAV flights? In order to do this, a DEM of difference method must be used (e.g. geomorphic change detection). With UAV photogrammetry you can obtain DSM, but you mention DTM, how did you convert DSM to DTM.
Response 4:Thank you for your specific comments.This paper focuses on the calculation and processing of field data. Detailed schemes have been made for field data collection, including the number of GCP and their layout scheme, the flight height and route of UAV, etc. In order to improve the accuracy of data calculation, GCP points are mainly measured by GNSS for multiple times, and the average value is taken as the final value. Calculate the RMSE of the DEM obtained in each period, subtract the data of the two periods to obtain the subsided basin data. Finally, the RMSE of subsidence basin was calculated by error propagation law, and then the accuracy of subsidence data was analyzed. UAV data obtained in each phase are filtered to remove noise points outside the terrain, and only the terrain information is retained. DSM is finally converted into DTM.
Point 5: Regarding the last remark, an orthophoto or satellite image of the area of interest would be useful. Was there vegetation present (which can be a problem for both UAV and InSAR)?
Response 5:Thank you for your specific comments.In this paper, UAV and InSAR are used to collect data in the mining area of western China. The less vegetation affected by climate has little influence on the accuracy of data solution.
Point 6:At R502 you mention the DTM of SRTM, but that has a very low resolution (approx. 30m) and can not be used in monitoring deformations.
Response 6:Thank you for your specific comments.Here is the initial elevation value of the research area set for the calculation of UAV data, instead of DEM of SRTM for monitoring subsidence. The final settlement monitoring and calculation are carried out with the results of the solution.
Point 7:The fusion can be further explained in the manuscript.
Response 7:Thank you for your specific comments.The specific fusion method is elaborated in Section 2.5.
Point 8:Discussion chapter needs improvements, with citations to similar studies
Response 8:Thank you for your specific comments.The Introduction has been modified and the corrections have been highlighted in red.

Round 2
Reviewer 3 Report
The revisions are small and many issues were not addressed, as well as evasive responses.
While there are positive aspects, there still are some issues and lacking aspects in the manuscript.
I will let the Academic Editor decide.